# PRIMT: Preference-based Reinforcement Learning with Multimodal Feedback and Trajectory Synthesis from Foundation Models

**Ruiqi Wang**[1]*  **Dezhong Zhao**[1,2]*  **Ziqin Yuan**[1]*  **Tianyu Shao**[1]
**Guohua Chen**[2]  **Dominic Kao**[1,3]  **Sungeun Hong**[4]  **Byung-Cheol Min**[1,5]
[1]Purdue University, West Lafayette, IN, USA
[2]Beijing University of Chemical Technology, Beijing, China
[3]University of Illinois Urbana-Champaign, Champaign, IL, USA
[4]Sungkyunkwan University, Seoul, South Korea
[5]Indiana University Bloomington, Bloomington, IN, USA

## Abstract

Preference-based reinforcement learning (PbRL) has emerged as a promising paradigm for teaching robots complex behaviors without reward engineering. However, its effectiveness is often limited by two critical challenges: the reliance on extensive human input and the inherent difficulties in resolving query ambiguity and credit assignment during reward learning. In this paper, we introduce PRIMT, a PbRL framework designed to overcome these challenges by leveraging foundation models (FMs) for multimodal synthetic feedback and trajectory synthesis. Unlike prior approaches that rely on single-modality FM evaluations, PRIMT employs a hierarchical neuro-symbolic fusion strategy, integrating the complementary strengths of large language models and vision-language models in evaluating robot behaviors for more reliable and comprehensive feedback. PRIMT also incorporates foresight trajectory generation, which reduces early-stage query ambiguity by warm-starting the trajectory buffer with bootstrapped samples, and hindsight trajectory augmentation, which enables counterfactual reasoning with a causal auxiliary loss to improve credit assignment. We evaluate PRIMT on 2 locomotion and 6 manipulation tasks on various benchmarks, demonstrating superior performance over FM-based and scripted baselines. Website at `https://primt25.github.io/`.

## 1 Introduction

Reinforcement learning (RL) has shown great success in various robotics domains [1–4], yet it remains reliant on carefully designed reward functions. In many practical scenarios, designing an informative reward function is highly challenging, as task objectives are often implicit and multi-faceted [5]. Preference-based RL (PbRL) [6–8] has emerged as a promising alternative to address this challenge by learning reward models from human comparative feedback over robot trajectories, providing a more intuitive means of aligning robotic systems with human intent [9–12]. Nevertheless, the extensive human input required for preference labeling restricts the scalability of PbRL [13].

To mitigate this bottleneck, recent work has explored leveraging foundation models (FMs), e.g., large language models (LLMs) and vision-language models (VLMs), as synthetic feedback sources, drawing on their broad world knowledge [14–17]. Compared to using FMs to design dense reward functions [18–20] or provide auxiliary contrastive signals [21–24], incorporating them as evaluators within PbRL offers a potentially more efficient and robust paradigm.

---

*Equal contribution. Corresponding authors: `wang5357@purdue.edu`; `minb@iu.edu`

39th Conference on Neural Information Processing Systems (NeurIPS 2025).

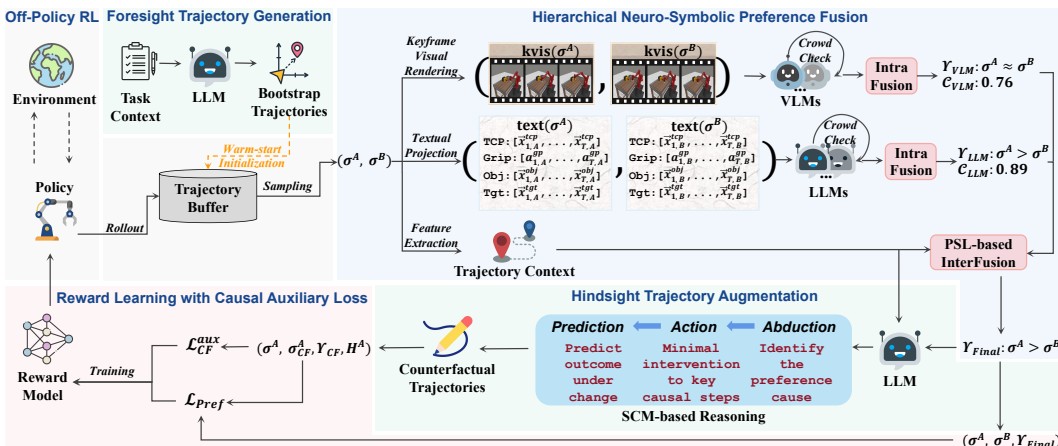

Figure 1: Overview of PRIMT, comprising two synergistic modules: 1) Hierarchical neuro-symbolic preference fusion improves the quality of synthetic feedback by leveraging the complementary strengths of VLMs and LLMs for multimodal evaluation of robot behaviors; and 2) Bidirectional trajectory synthesis mitigates early-stage query ambiguity through foresight generation and enhances credit assignment in reward learning via hindsight counterfactual augmentation with a causal auxiliary loss.

However, obtaining reliable and high-quality FM feedback remains challenging, primarily due to the dominant reliance on *single-modality* evaluation. LLM-based approaches [15, 16] interpret structured textual projections of trajectories, such as sequential arrays of state-action pairs, enabling sophisticated temporal reasoning over procedural logic and motion progression [25, 26]. However, these textual descriptions can be abstract or incomplete, making LLMs prone to hallucinations of key events, especially when inferring fine-grained spatial interactions [27, 28]. On the other hand, VLM-based methods [14] analyze spatial cues from visual renderings of robot trajectories, such as final-state images or intermediate frames, effectively capturing spatial goal completion [29]. Yet these methods often overlook subtle temporal dynamics within the trajectory [30, 31]. Consequently, relying on either modality alone risks incomplete or unreliable feedback (see Appendix A for more analysis), highlighting the need for a more comprehensive multimodal evaluation framework.

Furthermore, even if feedback from FMs reaches human-expert-level quality, PbRL still faces two intrinsic challenges: i) Query ambiguity: trajectory pairs often exhibit uniformly low quality in early training stages. This happens when they are generated from random or weakly optimized policies and lack task-relevant variations, making it hard to elicit meaningful preferences [7, 32]; and ii) Credit assignment: even when reliable *trajectory-level* preferences are available, it often remains difficult to accurately attribute the observed preference differences to specific states or actions [33–35]. Without effective *state-action-level* credit assignment, the learned reward model may result in misaligned behaviors in subsequent RL training [33]. Meanwhile, FMs have shown strong abilities in planning [36, 37], control [38, 39], and causal reasoning [40, 41]. These advances lead us to the following question: *Can FMs move beyond serving as passive preference providers to be actively leveraged to mitigate query ambiguity and improve credit assignment in PbRL?*

In this paper, we propose PRIMT, a foundation model–driven framework for PbRL designed to address key challenges in synthetic feedback quality, query ambiguity, and credit assignment. PRIMT, as illustrated in Fig. 1, comprises two core components:

i) *Multimodal feedback fusion*, which enhances synthetic feedback quality by combining the complementary advantages of LLM- and VLM-based evaluations. Rather than directly feeding multimodal trajectory representations into generic multimodal FMs, PRIMT adopts a hierarchical neuro-symbolic preference fusion strategy. It first performs intra-modal fusion to produce modality-specific labels and confidence estimations. Then, inter-modal fusion is conducted using probabilistic soft logic (PSL) [42], which infers the final preference label via structured and interpretable reasoning over multimodal evaluation outputs and trajectory context, enabling robust aggregation of preference beliefs from heterogeneous sources.

ii) *Bidirectional trajectory synthesis*, which leverages FMs to actively enhance reward learning in PbRL. In the foresight phase, LLMs generate diverse task-aligned trajectories to initialize the

trajectory buffer. Unlike prior work [16, 36] that assumes FM-generated trajectories are optimal, we treat them as semantically meaningful anchors for informative comparisons, reducing early-stage query ambiguity. In the hindsight phase, LLMs are prompted to generate counterfactual trajectories via causal reasoning based on the structural causal model (SCM) [43]. When a clear preference is detected, the model identifies causal steps and applies minimal edits to reverse the preference, producing counterfactuals that highlight critical distinctions. To better exploit these samples for credit assignment, we introduce a causal-aware auxiliary loss that enforces reward separation at edited steps while ensuring consistency elsewhere. This enables more precise preference attribution, thereby improving the efficiency and generalization of the learned reward model in downstream RL training.

Our key contributions are summarized as follows:

- We present PRIMT, a general FM-driven framework for zero-shot PbRL, which leverages foundation models not only as synthetic teachers to eliminate human annotation but also as active agents to facilitate preference reward learning.
- We introduce a hierarchical neuro-symbolic preference fusion strategy that combines the complementary strengths of LLMs and VLMs for multimodal evaluation of robot trajectories, improving the reliability and quality of synthetic feedback.
- We propose foresight trajectory generation to bootstrap early-stage query informativeness, and hindsight trajectory augmentation via counterfactual reasoning, coupled with a causal auxiliary loss, to improve credit assignment in reward learning.
- We conduct extensive experiments across 2 locomotion and 6 manipulation tasks from the DMC [44], MetaWorld [45], and ManiSkill [46] benchmarks, demonstrating that PRIMT consistently outperforms state-of-the-art baselines. Ablation studies provide insight into component effectiveness, and we further validate PRIMT's real-world applicability on a Kinova Jaco robot.

## 2  Related Works and Preliminaries

**Foundation Models as Rewards for RL**   Foundation models refer to large-scale pre-trained models with strong generalization and reasoning capabilities across tasks [47]. Recent work has explored leveraging FMs to address the reward engineering challenge in RL. One line of research uses coding LLMs to directly generate executable reward functions [18–20]. Another employs VLMs as contrastive reward signals [21–24]; for example, RoboCLIP [24] rewards agents by aligning trajectory images with task descriptions or demonstrations. However, such explicit FM-based reward signals are often noisy and high-variance [14]. Recent approaches have adopted the PbRL paradigm, using FMs as synthetic evaluators to generate trajectory-level preference labels and train reward models: PrefCLM [15] and RL-SaLLM-F [16] leverage LLMs to analyze numerical state-action sequences, while RL-VLM-F [14] uses VLMs to assess final-state images of robot trajectories. These approaches have shown improved performance over FM-generated scalar rewards. Our work builds on this PbRL-with-FM direction but introduces two key innovations. First, PRIMT adopts a multimodal evaluation scheme that combines VLM and LLM perspectives via hierarchical neuro-symbolic fusion, improving the robustness and quality of synthetic supervision. Second, rather than using FMs solely for passive evaluation, we actively incorporate them to facilitate reward learning via trajectory synthesis.

**Preference-based RL**   PbRL aims to learn a reward model $r_\psi$ from human comparative feedback over pairs of robot trajectories [7]. A trajectory $\sigma$ is defined as a sequence of states and actions $\{(s_1, a_1) \dots, (s_T, a_T)\}$ with a length of $T$. The annotator provides a preference label $\Upsilon \in \{-1, 0, 1\}$ for each pair $(\sigma^A, \sigma^B)$, where $\Upsilon = 1$ indicates that $\sigma^A$ is preferred, 0 means $\sigma^B$ is preferred, and $-1$ denotes indecision. A preference predictor is constructed using the Bradley-Terry model [48] to estimate the preference probabilities. The likelihood that $\sigma^A$ is preferred over $\sigma^B$ is computed as:

$$P_\psi[\sigma^A \succ \sigma^B] = \frac{\exp\left(\sum_{t=1}^T r_\psi(s_t^A, a_t^A)\right)}{\exp\left(\sum_{t=1}^T r_\psi(s_t^A, a_t^A)\right) + \exp\left(\sum_{t=1}^T r_\psi(s_t^B, a_t^B)\right)} \tag{1}$$

The reward model is trained to align with human preferences by minimizing a cross-entropy objective over a collected preference dataset $\mathcal{D} = \{(\sigma^A, \sigma^B, \Upsilon)\}$ as:

$$\mathcal{L}_{\text{Pref}} = -\mathbb{E}_{(\sigma^A, \sigma^B, \Upsilon) \sim \mathcal{D}} \left[ \mathbb{I}\{\Upsilon = 1\} \log P_\psi[\sigma^B \succ \sigma^A] + \mathbb{I}\{\Upsilon = 0\} \log P_\psi[\sigma^A \succ \sigma^B] \right] \tag{2}$$

Training alternates between reward learning and RL-based policy optimization with learned reward.

**Query Ambiguity**   PbRL relies on informative preference queries to train effective reward models. However, trajectory pairs often exhibit low task-relevant diversity, leading to query ambiguity [8]. This ambiguity is especially pronounced in early training stages, when trajectories are uniformly low-quality and incoherent due to randomly initialized policies [49]. Prior works address this by selecting maximally distinguishable or uncertain samples [8, 32, 50] or initializing the reward model with expert demonstrations [51, 52]. More recent works employ LLMs to revise ambiguous trajectories during training, assuming that the edited versions are task-complete and preferred [16]. In contrast, we proactively initialize the replay buffer with LLM-generated trajectories that are diverse and task-aligned. Unlike prior work, we do not assume these are optimal, but use them as preference anchors to support more informative and efficient early-stage evaluation.

**Credit Assignment**   Another core challenge in PbRL is the granularity mismatch between trajectory-level preference supervision and the desired state-action-level reward signal [33]. This mismatch introduces uncertainty in attributing credit to specific decisions, impairing both the alignment and generalization of the learned reward model [53]. Prior work mitigates this issue by training transformer-based world models that estimate state importance [33, 54, 55], or by collecting additional human annotations to highlight key moments [34]. In contrast, our approach requires neither extra supervision nor architectural changes. Inspired by causal counterfactual reasoning [43, 56–58], we prompt LLMs to generate hindsight-based counterfactual trajectories by minimally editing key decision points in the preferred trajectory to reverse the preference. By asking, *"What minimal change would make this trajectory less preferred?"*, we obtain contrastive examples that expose the underlying reasons for preference. To effectively leverage these counterfactuals in reward learning, we introduce a causal-aware auxiliary loss. It enforces reward separation at edited points while maintaining consistency in the unedited parts, leading to more precise credit assignment.

## 3   Methodology

In this section, we present PRIMT: **PR**eference-based re**I**nforcement learning with **M**ultimodal feedback and **T**rajectory synthesis from foundation models. An overview of the PRIMT is illustrated in Fig. 1. Detailed prompts with example outputs for each component are included in Appendix C.

### 3.1   Multimodal Feedback Generation and Fusion

**Trajectory Preprocessing**   Given a trajectory pair $(\sigma^A, \sigma^B)$ sampled from the trajectory buffer, we first obtain their textual projections $text(\sigma^A)$ and $text(\sigma^B)$ for LLM-based evaluation, following [16]. These projections organize each trajectory into dimension-specific sequences, capturing structured temporal patterns across state and action components in a format that enhances semantic interpretability. For VLM-based evaluation, instead of using all frames or final-state images as in prior work [14], we propose a hybrid keyframe extraction method to capture both low-level motion cues and high-level behavior transitions while avoiding visual overload: i) near-zero velocity detection identifies frames where the robot motion is minimal, typically marking subgoal completions or transitional pauses [59]; ii) smoothing residual peaks detect high-curvature or abrupt motion transitions by comparing the raw trajectory to its smoothed version, capturing key motion shifts [60]; and iii) change point detection segments the trajectory into semantically coherent phases to identify structural high-level task changes [61]. We take the union of the selected frames from all three methods, together with the first and last steps of the trajectory, to form the final keyframe sets $kvis(\sigma^A)$ and $kvis(\sigma^B)$. Full details are provided in Appendix B.

**Intra-modal Preference Fusion**   We then query LLM and VLM separately with corresponding textual projections and keyframe sequences, along with a brief task description, to elicit preference judgments. Each query follows a structured three-step chain-of-thought (CoT) prompt: i) analyze each trajectory in terms of its effectiveness in achieving the task goal; ii) based on this analysis, output a preference label; and iii) verify the decision and assign a confidence score from $0$ to $1$, reflecting the preference certainty. To mitigate variance and improve the reliability of intra-modal labels, we adopt a crowd-check mechanism, querying LLM or VLM multiple times with randomly permuted trajectory orderings. This produces $K$ predictions from each feedback modality $M \in \{\text{LLM}, \text{VLM}\}$, each consisting of a preference label $\Upsilon_M^{(k)} \in \{-1, 0, 1\}$ and a confidence score $\mathcal{C}_M^{(k)} \in [0, 1]$. We then

aggregate these judgments into a final modality-specific preference label $\Upsilon_M$ via major voting as:

$$\Upsilon_M = \underset{l \in \{-1,0,1\}}{\operatorname{argmax}} \sum_{k=1}^{K} \mathbb{I}(\Upsilon_M^{(k)} = l) \tag{3}$$

To estimate and calibrate the confidence $\mathcal{C}_M$ associated with the final label, we compute a weighted combination of two complementary signals: i) the average confidence $\bar{\mathcal{C}}_M$ among $N$ judgments that agree with the final label, and ii) the label consistency ratio $\dot{\mathcal{C}}_M$ representing vote agreement:

$$\bar{\mathcal{C}}_M = \frac{1}{N} \sum_{k=1}^{K} \mathcal{C}_M^{(k)} \cdot \mathbb{I}(\Upsilon_M^{(k)} = \Upsilon_M); \;\; \dot{\mathcal{C}}_M = \frac{1}{K} \sum_{k=1}^{K} \mathbb{I}(\Upsilon_M^{(k)} = \Upsilon_M) \tag{4}$$

The final confidence $\mathcal{C}_M$ is then computed as:

$$\mathcal{C}_M = \alpha \cdot \bar{\mathcal{C}}_M + (1 - \alpha) \cdot \dot{\mathcal{C}}_M \tag{5}$$

where $\alpha \in [0, 1]$ is a balancing hyperparameter (typically set to 0.5). This formulation ensures that the final confidence reflects both internal certainty and stability under input perturbations, thereby improving the robustness of modality-specific confidence estimation.

**Inter-modal Preference Fusion**   The next step is to integrate modality-specific preference labels into a unified decision. This process is non-trivial, as it must consider multiple factors, including intra-modal uncertainty, cross-modal conflicts, and trajectory context that reflects the relative difficulty of visual versus textual evaluation. Intuitively, one might define heuristic rules for each factor, for example, favoring the VLM label when the visual difference between trajectories is high, or trusting the label with higher confidence. Yet, such heuristics are brittle and hard to generalize: the conditions involved are often continuous rather than binary, and the interactions among rules can be complex.

To efficiently model these latent dependency structures among inputs, heuristics, and decisions, we employ Probabilistic Soft Logic (PSL) [42], a probabilistic framework representing entities of interest as logical *atoms* interconnected by weighted first-order logic *rules*. Specifically, we define four rules to guide inter-modal preference fusion:

***i) Agreement Rule***: If the VLM and LLM agree on the same preference label $\Upsilon$ and at least one modality reports high confidence, the agreed label is used as the final decision:

$$\forall \Upsilon, M : \texttt{IsAgree}(\Upsilon) \wedge \texttt{ConfHigh}(M) \rightarrow \texttt{FinalLabel}(\Upsilon) \tag{6}$$

Here, $\texttt{IsAgree}(\Upsilon)$ is a binary indicator set to 1 if both VLM and LLM predict the same label $\Upsilon$, and 0 otherwise; $\texttt{ConfHigh}(M)$ is a continuous atom representing the modality-specific confidence score $\mathcal{C}_M \in [0, 1]$; and $\texttt{FinalLabel}(\Upsilon) \in [0, 1]$ is the output atom to be inferred by PSL, representing the final soft confidence assigned to label $\Upsilon \in \{-1, 0, 1\}$.

***ii) Conflict Resolution Rules***: When modality-specific labels conflict, we resolve the disagreement by considering the associated confidence and trajectory context (detailed rationale can be found in Appendix A.2). Specifically, we prioritize the VLM prediction if the visual discriminability between trajectories and VLM confidence is high:

$$\forall \Upsilon : \neg \texttt{IsAgree}(\Upsilon) \wedge \texttt{VLMLabel}(\Upsilon) \wedge \texttt{ConfHigh}(\text{VLM}) \wedge \texttt{VDHigh} \wedge \rightarrow \texttt{FinalLabel}(\Upsilon) \tag{7}$$

Likewise, if the LLM predicts a label $\Upsilon$ with high confidence and the temporal discriminability of the trajectory pair is high, we prioritize the LLM prediction:

$$\forall \Upsilon : \neg \texttt{IsAgree}(\Upsilon) \wedge \texttt{LLMLabel}(\Upsilon) \wedge \texttt{ConfHigh}(\text{LLM}) \wedge \texttt{TDHigh} \rightarrow \texttt{FinalLabel}(\Upsilon) \tag{8}$$

Here, $\texttt{VLMLabel}(\Upsilon)$ and $\texttt{LLMLabel}(\Upsilon)$ are indicators set to 1 if the modality predicts label $\Upsilon$, and 0 otherwise. The atom $\texttt{VDHigh}$ captures the visual discriminability between the two trajectories as:

$$\texttt{VDHigh} = \rho \left( \mathcal{W}(f(kvis(\sigma^A)), f(kvis(\sigma^B))) \right) \tag{9}$$

where $f(\cdot)$ denotes the CLIP encoder applied to keyframe sets $kvis(\cdot)$, $\mathcal{W}$ denotes the Wasserstein distance, and $\rho(\cdot)$ is a sigmoid function used for normalization.

Similarly, $\texttt{TDHigh}$ captures temporal discriminability based on trajectory volatility differences:

$$\texttt{TDHigh} := \rho \left( \left| \mathrm{TrjVol}(\sigma^A) - \mathrm{TrjVol}(\sigma^B) \right| \right) \tag{10}$$

where $\mathrm{TrjVol}(\cdot)$ measures the state-action volatility of a trajectory, defined as the mean $L2$ norm of second-order finite differences:

$$\mathrm{TrjVol}(\sigma) = \frac{1}{T-2} \sum_{t=2}^{T-1} \|(s_{t+1}, a_{t+1}) - 2(s_t, a_t) + (s_{t-1}, a_{t-1})\|_2.$$

(11)

***iii) Indecision Rule***: When both modalities exhibit low confidence, we assign the indecision label:

$$\neg\texttt{ConfHigh}(\texttt{VLM}) \wedge \neg\texttt{ConfHigh}(\texttt{LLM}) \rightarrow \texttt{FinalLabel}(-1)$$

(12)

During PSL inference, each logical atom is instantiated with data and grounded into either an observed input variable $X$ (e.g., `IsAgree`, `ConfHigh`, `TDHigh`) or an output variable $Y$ (e.g., `FinalLabel`). Valid substitutions of these atoms within rule templates yield a set of ground rules. Each ground rule induces one or more hinge-loss potentials, relaxed from the logical clauses using Łukasiewicz continuous-valued semantics. Formally, each potential takes the form:

$$\phi(Y, X) = [\max(0, \ell(Y, X))]^p$$

(13)

where $\ell$ is a linear function in PSL representing the distance to satisfaction of the corresponding ground rule, and $p \in \{1, 2\}$ controls whether the penalty is linear or quadratic. Given observed variables $X$ and target variables $Y$, PSL defines a hinge-loss Markov random field and performs inference by solving a convex constrained optimization problem (more details of PSL inference are provided in Appendix D):

$$Y^* = \arg\min_Y \sum_{i=1}^m w_i, \phi_i(Y, X) \quad \text{s.t.} \sum_{\Upsilon \in \{-1, 0, 1\}} \texttt{FinalLabel}(\Upsilon) = 1$$

(14)

where $m$ is the number of instantiated potentials, $\phi_i$ denotes the $i^{\text{th}}$ potential function, and $w_i$ is the weight assigned to the corresponding rule template. Unlike standard PSL formulations, we impose a one-hot constraint over the final label atoms to reflect the single-label nature in PbRL. By encoding structured dependencies among modality-specific outputs and trajectory-level context, PSL facilitates robust and adaptive integration of complementary cues from multiple feedback modalities, effectively managing uncertainties and cross-modal conflicts.

## 3.2 Bidirectional Trajectory Synthesis

**Foresight Trajectory Generation**    Prior to PbRL training, we employ LLMs to generate bootstrapped trajectories that exhibit diverse, semantically meaningful, and task-aligned behaviors, providing a warm-start initialization for the trajectory buffer. Inspired by structured code-generation paradigms [36, 38], we adopt a three-step CoT strategy: i) generate a high-level, multi-step action plan from the task specification; ii) translate each step into executable code snippets that implement concrete motion primitives; and iii) execute these programs under varied initial conditions (e.g., robot start positions) and strategy parameters (e.g., height to approach the target) to synthesize a diverse set of plausible trajectories. Compared to directly prompting LLMs to generate low-level trajectory arrays [16], our method improves physical feasibility and semantic coherence by grounding trajectory synthesis in program logic. The generated trajectories are considered as bootstrapped demonstrations rather than optimal ones, subsequently evaluated by our multimodal feedback module. Combined with strategic sampling schemes, such as uncertainty-based sampling [8], these trajectories serve as informative preference anchors when paired with exploration trajectories, reducing ambiguity in early-stage preference queries and accelerating reward learning.

**Hindsight Trajectory Augmentation with Causal Auxiliary Loss**    During PbRL training, whenever a clear preference is identified by the multimodal feedback module, we prompt LLMs to perform hindsight reasoning to generate counterfactual variants of the preferred trajectory. This process follows a three-step reasoning pattern based on the structural causal model [43]:

***i) Abduction:*** Identify the causal rationale behind the observed preference by extracting a set of critical causal steps $T^*$ in the preferred trajectory $\sigma^*$ that contribute to the preference. To assist this process, we provide the step indices corresponding to keyframes in $kvis(\sigma^*)$ as reference candidates, though the selected causal steps are not necessarily limited to them.

*ii) Action:* Select a key step $t^* \in T^*$ for minimal intervention, generating a counterfactual trajectory $\sigma^*_{cf}$ that reverses the original preference. This involves modifying some critical state-action features at the selected step, such as introducing a small gripper delay or adding a local end-effector position perturbation. The rest of the trajectory remains identical to the original while we allow the LLMs to apply light smoothing to the immediate neighbors (e.g., 2-3 steps before and after the intervention) to ensure physical continuity and avoid abrupt state transitions. Multiple counterfactual variants can be generated through repeated LLM sampling, providing a diverse set of sub-preferred alternatives. Following the minimal edit principle [62, 63], we filter the generated counterfactuals based on the L1 distance between the edited state-action pairs and the original, ensuring a small deviation threshold.

*iii) Prediction:* Pair each counterfactual variant with the originally preferred trajectory and feed them into the LLM-based intra-modal fusion module to verify whether the counterfactual is sub-preferred, i.e., satisfying the preference condition ($\sigma^* \succ \sigma^*_{cf}$). Only counterfactuals that meet this criterion are stored and used for reward learning.

Through this hindsight reasoning process, we now have counterfactual trajectories of the preferred trajectory that share a common structure except at minimally edited steps, because of which their preference outcomes diverge. As such, we can assume that the edited steps are responsible for the observed preference signal, which the reward model should learn to correctly attribute. Given this, we introduce a causal auxiliary loss that encourages discriminability at the edited steps while maintaining consistency elsewhere to guide the model to focus on causal differences that drive preferences:

$$\mathcal{L}^{\text{aux}}_{\text{cf}} = \underbrace{\sum_{t=1}^{T} H_t \cdot \log\left(1 + \exp\left(r_\psi(s_t^{cf}) - r_\psi(s_t^*)\right)\right)}_{\text{i) causal contrast loss}} + \underbrace{\sum_{t=1}^{T} (1 - H_t) \cdot \left\| r_\psi(s_t^*) - r_\psi(s_t^{cf}) \right\|_2^2}_{\text{ii) reward consistency loss}} \quad (15)$$

where $H_t$ is a binary mask indicating the edited steps, i.e., $H_t = 1$ for edited time steps and $H_t = 0$ otherwise. The first term encourages the model to assign higher rewards to the preferred trajectory at casual steps, while the second enforces consistent rewards on unchanged regions. This auxiliary loss is combined with the trajectory-level preference loss as in Eq. 2, forming the final loss for reward learning with the generated counterfactuals:

$$\mathcal{L}_{\text{final}} = \mathcal{L}_{\text{pref}} + \lambda_{\text{cf}} \cdot \mathcal{L}^{\text{aux}}_{\text{cf}} \quad (16)$$

where $\lambda_{\text{cf}}$ is a weight used to scale the auxiliary loss to the same magnitude as the primary preference loss. This integrated loss formulation enables the reward model to capture trajectory-level preferences while providing more precise state-action-level credit attributions.

## 4 Experiments

### 4.1 Setup

We evaluate PRIMT across a diverse set of tasks, including 2 locomotion tasks: *Hopper Stand* and *Walker Walk* from DeepMind Control (DMC) suite [44], as well as 6 articulation or rigid body manipulation tasks: *Button Press*, *Door Open*, and *Sweep Into* from MetaWorld suite [45], and *PickSingleYCB*, *StackCube*, and *PegInsertionSide* from ManiSkill suite [46]. Detailed task descriptions are provided in Appendix E. We compare PRIMT against the following baselines:

- **RL-VLM-F [14]:** This baseline utilizes VLM to analyze visual renderings of trajectories to provide preference labels, representing a state-of-the-art VLM-based method.

- **RL-SaLLM-F [16]:** This model employs LLM to provide preference labels based on textual descriptions of trajectories, and to modify ambiguous trajectories by generating self-improved alternatives, which are assumed to be preferred when paired with the original ones. This denotes a LLM-based method with trajectory augmentation for addressing query ambiguity.

- **PrefCLM [15]:** This baseline leverages crowdsourced LLMs to provide evaluation feedback for improved feedback quality, presenting another state-of-the-art LLM-based method.

- **PrefMul:** We build this baseline by directly providing the multimodal trajectory inputs used in PRIMT to a multimodal FM for evaluation, representing a naive approach to multimodal feedback.

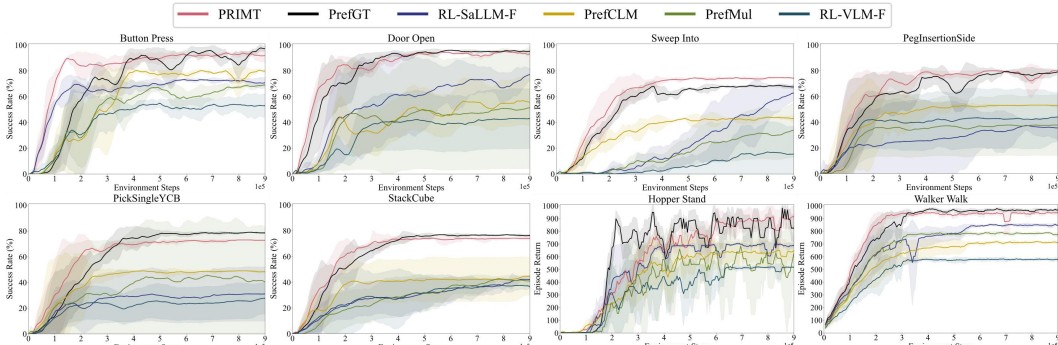

Figure 2: Learning curves of PRIMT and baseline methods across all tasks, averaged over 5 runs with solid lines denoting the average and shaded regions representing the standard error. A moving average window of 5 steps for locomotion tasks and 10 steps for manipulation tasks is applied to improve readability.

- **PrefGT:** This baseline uses expert-designed reward functions provided by the benchmarks in a scripted teacher manner [8] to provide preference labels. This should, in theory, serve as an upper-bound oracle of PbRL performance on each task.

We also build several ablation models to assess the impact of each PRIMT component:

- **w/o.Intra:** without the crowd-check mechanism and intra-modal preference fusion module.
- **w/o.Inter:** without the inter-modal preference fusion module, directly selecting the modality-specific label with highest confidence as the final label.
- **w/o.ForeGen:** without the foresight trajectory generation module.
- **w/o.HindAug:** without the hindsight trajectory augmentation module.
- **w/o.CauAux:** without the causal auxiliary loss for counterfactuals in reward learning.

To eliminate the impact of non-model factors, we use the same trajectory inputs and CoT prompts as in PRIMT for preference label elicitation across all FM-based baselines and ablation models. The only exception is PrefCLM, which relies heavily on direct access to the environment code; for this baseline, we follow the original settings from the source paper. For all FM-based methods, we use `gpt-4o` as the LLM backbone. For the PbRL backbone, we use PEBBLE [7] with the uncertainty-based sampling schedule [8] for all methods, along with a consistent set of hyperparameters for the RL-based policy learning phase with SAC. This design ensures that the only difference between methods lies in the preference reward learning, allowing for a more controlled comparison.

We evaluate all baselines across all tasks and conduct ablation studies on the *Door Open* and *PickSingleYCB* tasks, each with five random seed runs to ensure statistical robustness. For manipulation tasks, we report the success rate, whereas for locomotion tasks, we use the episodic return provided by the benchmarks. Further implementation details are provided in Appendix F.

### 4.2 Does PRIMT Learn Effective Rewards and Policies that Prompt Task Performance?

We first investigate whether PRIMT can learn effective reward models that lead to policies capable of solving complex tasks. Fig. 2 shows the learning curves of all methods across 8 tasks. We observe that PRIMT consistently outperforms all FM-based baselines that rely on single-modality feedback, demonstrating superior final performance and faster convergence. However, the naive multi-

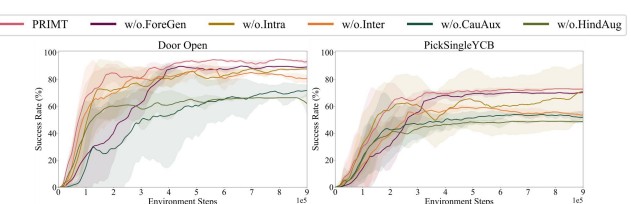

Figure 3: Learning curves of PRIMT and ablation models on the *Door Open* and *PickSingleYCB* tasks.

modal feedback method, PrefMul, performs poorly. We attribute this to the fact that directly feeding multimodal trajectory inputs to multimodal FMs without appropriate fusion can overwhelm the

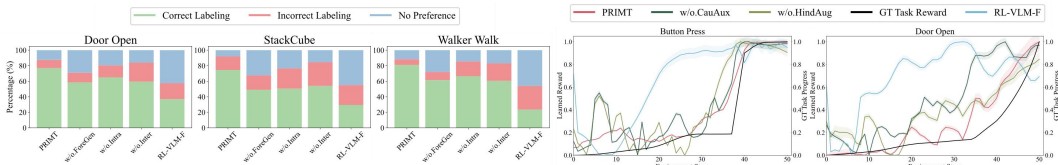

Figure 4: **Left:** Distribution of preference labels, showing the proportion of correct, incorrect, and indecisive labels across different methods. **Right:** Reward alignment analysis, comparing the learned reward outputs of PRIMT, ablations, and baselines against ground-truth rewards. Plots on more tasks are provided in Appendix G.1.

models, potentially even degrading performance. This highlights the need for carefully designed fusion strategies to fully leverage the complementary strengths of multimodal signals, as implemented in the hierarchical neuro-symbolic fusion strategy of PRIMT. Moreover, each component in this hierarchical design is crucial: as shown in Fig. 3, removing either intra-modal or inter-modal fusion significantly degrades task performance. This drop is especially severe when inter-modal fusion is removed, underscoring the critical role of PSL-based reasoning in PRIMT. Unlike simple rule-based fusion in w/o.Inter, PSL-based fusion more effectively captures intra-modal uncertainty, cross-modal conflicts, and the influence of trajectory context, enabling more robust preference integration from heterogeneous sources.

Notably, PRIMT is competitive with the oracle PrefGT, even surpassing it on the *Sweep Into* and *Peg Insertion Side* tasks, and generally achieving faster early-stage learning, while other FM-based methods fall behind. This suggests that while PrefGT benefits from fine-grained oracle preference labels, PRIMT narrows this gap by improving the quality of synthetic feedback and leveraging trajectory synthesis to address inherent challenges in PbRL. As shown in Fig. 3, foresight generation appears to contribute more to the acceleration of early-stage learning, as the w/o.ForeGen variant reaches a similar final performance but learns more slowly in the initial stages. In contrast, hindsight augmentation plays a critical role in achieving high final performance, as evidenced by the significant performance drop when either w/o.HindAug or w/o.CauAux is removed. Interestingly, while w/o.CauAux can still benefit from the counterfactual trajectory augmentation, its performance is significantly worse, indicating that the causal auxiliary loss we designed can more effectively leverage these counterfactuals in reward learning, capturing state-action-level preference causation.

### 4.3 Does PRIMT Improve the Quality of Synthetic Feedback and Mitigate Query Ambiguity?

We next examine the distribution of synthetic feedback generated by PRIMT to assess its impact on preference label quality and query informativeness. We calculate accuracy by comparing the synthetic preference labels with those in PrefGT and record the preference decisions. Fig. 4 shows the percentages of correct labeling, incorrect labeling, and preference indecision for PRIMT, w/o.ForeGen, w/o.Intra, w/o.Inter, and RL-VLM-F (left to right). We observe that PRIMT consistently produces more accurate preference labels compared to the baseline and ablations, confirming the effectiveness of the hierarchical fusion design in improving label quality. Additionally, PRIMT significantly reduces indecision rates compared to both the baseline and w/o.ForeGen, indicating that foresight generation effectively mitigates query ambiguity.

It is worth noting that we could not directly compare indecision rates with RL-SaLLM-F, another method that uses LLMs to address query ambiguity, because it inherently eliminates indecision by always treating self-augmented trajectories as preferable. However, as shown in Fig. 2, this baseline struggles with early-stage learning on tasks from the ManiSkill, which involve high-dimensional state and action spaces. We attribute this to the difficulty of directly generating optimal trajectories at the low level in such tasks, making the assumption that generated trajectories are always preferable highly misleading. This highlights the advantages of our foresight generation approach, which addresses query ambiguity from the outset by initializing diverse, bootstrapped trajectories as potential preference anchors, and our code-generation paradigm, which improves the trajectory sample quality.

### 4.4 Does PRIMT Enhance Credit Assignment in the Reward Model?

We further investigate how the learned reward models align with the task progress at state-action level. Fig. 4 shows the normalized reward outputs from the learned reward models of PRIMT, its two

ablations without full hindsight trajectory augmentation, and the baseline RL-VLM-F, along with the normalized ground-truth reward values on the same trajectories. We observe that PRIMT produces more aligned reward patterns that closely reflect task progress, while the baselines and ablations exhibit either noisy signals or high variance, indicating that their learned reward models struggle to accurately assign rewards at the state-action level, even if they capture trajectory-level preferences. This demonstrates that PRIMT, particularly its hindsight augmentation and causal auxiliary loss, enables more precise state-action-level credit assignment in reward learning. Extra quantitative results using the $R^2$ coefficient are provided in Appendix G.8.

### 4.5 Additional Experimental Results

We conducted additional analyses to provide further validation insights of the proposed framework. These include: qualitative evaluations of the trajectory synthesis module (Appendices G.3 and G.4), visualization of policy outcomes across different methods (Appendix G.5), comparison with the dense-reward RL baseline (Appendix G.7), and preliminary experiments on a more complex dual-arm manipulation task (Appendix G.6).

We also performed ablation studies on the influence of the foundation model backbone (Appendix G.2). The results show good potential for more accessible deployment with smaller or cheaper foundation models: with GPT-4o-mini, we achieved a 94% reduction in cost with a 16–26% drop in performance, leading to a 13× improvement in cost–performance efficiency.

### 4.6 Real-world Deployment

We further demonstrated the effectiveness of PRIMT on a Kinova Jaco robot in block lifting and stacking tasks (Fig. 5). Detailed experimental settings and results are provided in Appendix G.9.

**Block Lifting**  **Block Stacking**

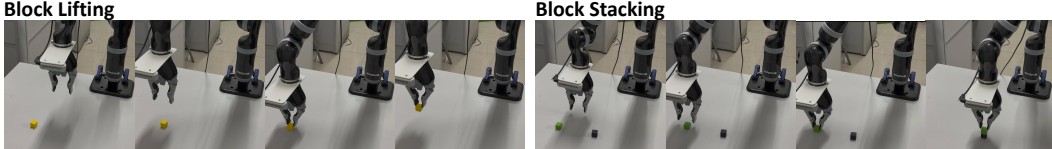

Figure 5: PRIMT successfully completes real-world block lifting and stacking tasks on a Kinova Jaco robot.

## 5 Conclusion and Future Work

In this work, we presented PRIMT, a method that leverages foundation models for multimodal synthetic evaluation and trajectory synthesis to reduce human effort and address query ambiguity and credit-assignment challenges in preference-based reinforcement learning (PbRL). We demonstrated the advantages of PRIMT across a wide range of locomotion and manipulation tasks, achieving superior task performance, higher-quality synthetic feedback, and more accurately aligned reward models.

A limitation of this study involves the increased cost associated with foundation-model usage due to the multimodal components of the framework. To provide a clearer understanding of the resource requirements, we conducted a detailed analysis of computational usage and the corresponding cost–performance trade-offs, presented in Appendix H.1. We observe that PRIMT strikes a good balance between performance and cost-effectiveness, providing a practical path toward scalable preference learning.

Looking forward, while PRIMT was evaluated in single-agent robotic domains, extending its key components, such as multimodal evaluation and trajectory synthesis, to non-robotic and multi-agent settings represents an exciting direction for future work. Further discussion of assumptions, limitations, and broader impacts is provided in Appendix H.

## Acknowledgments

The authors would like to thank Jinjin Cai from Purdue University for insightful comments on the figures and visualizations. This work was partially supported by the Ministry of Science and ICT (MSIT), Korea, under the Global Research Support Program in the Digital Field (RS-2024-00425354), supervised by the Institute for Information & Communications Technology Planning & Evaluation (IITP). The support applies to authors Hong and Min.

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

# APPENDIX

## Table of Contents

# A  Limitations of Single-Modality Feedback in Robot Trajectory Evaluation

In this section, we provide a detailed analysis of how single-modality evaluation patterns can lead to incomplete or unreliable feedback in robot trajectory judgment. We also present the rationale behind our hierarchical neuro-symbolic fusion module design, which addresses these limitations by integrating complementary signals from VLMs and LLMs.

## A.1  Failure Cases of VLM- and LLM-Based Preference Feedback

To illustrate the potential pitfalls of relying solely on VLM or LLM evaluations, we present a series of failure cases from the Door Open task in the MetaWorld benchmark. These cases highlight situations where either VLMs or LLMs struggle to provide accurate preference judgments, leading to suboptimal or inconsistent feedback. For each example, we use structured prompts and trajectory inputs as in PRIMT, with responses obtained by querying `gpt-4o` as the representative VLM/LLM.

**VLM Limitations and LLM Advantages**  Figs. 6a and 6b present two illustrative cases from the MetaWorld Door Open task, highlighting the complementary strengths of LLMs and the inherent limitations of VLMs in trajectory evaluation.

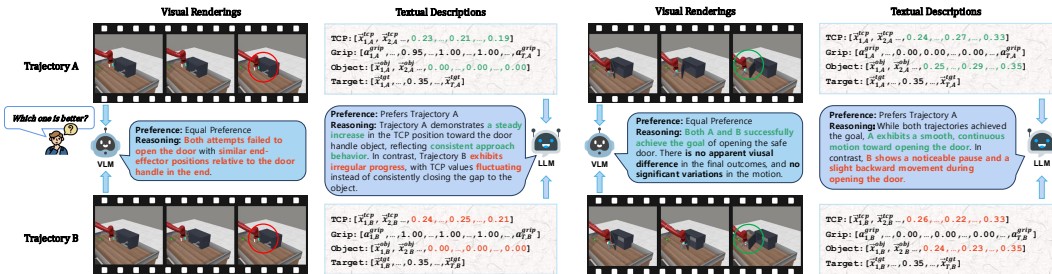

(a) LLM accurately infers task intent from TCP movement patterns, while VLM struggles to distinguish between visually similar failure cases.

(b) LLM captures motion fluency and continuous state transitions, while VLM is misled by similar frame appearances.

Figure 6: Examples where LLM-based evaluation outperforms VLM-based evaluation, highlighting the advantages of temporal reasoning and task intent understanding.

In Fig. 6a, both Trajectory A and Trajectory B ultimately fail to open the door, resulting in similar final visual states. The VLM, relying solely on frame-based image sequences, cannot differentiate the two and returns an indecisive response. In contrast, the LLM, which processes structured state encodings (e.g., TCP position, gripper state, object coordinates), explicitly favors Trajectory A. Its response indicates that Trajectory A exhibits a more consistent progression toward the door handle, reflecting clearer task intent, while Trajectory B lacks such directed movement. This distinction is critical for evaluating task success, as it captures the underlying purpose of each motion sequence rather than merely the final outcome.

Fig. 6b presents a different challenge, where both trajectories successfully complete the task. Despite similar final visual outcomes, the LLM identifies qualitative differences in the motion patterns, preferring Trajectory A for its smooth, uninterrupted approach. In contrast, Trajectory B exhibits a brief hesitation and slight backward drift, disrupting the task's fluidity. These subtle behavioral cues, emphasized by the green circle in Trajectory A and the red circle in Trajectory B, are detectable only through temporal reasoning and sequential context, which are inherently absent from frame-based VLM evaluations.

These examples underscore a fundamental limitation of VLMs: their reliance on discrete visual frames makes them highly effective at capturing spatial relationships but poorly suited for interpreting task intent, motion quality, or temporal consistency. Without structured state data, VLMs can be easily misled by visually similar, yet semantically different, trajectories, leading to indecision or incorrect preferences.

In contrast, LLMs offer a powerful complement by incorporating temporal context, semantic task cues, and structured state information into their evaluations. This allows them to assess not just where a robot ends up, but how it arrived there, capturing fine-grained distinctions like hesitation-free execution, consistent approach angles, and purposeful task progression. These strengths make LLMs an ideal complementary modality for evaluating complex robotic behaviors, particularly in tasks where motion fluency and goal-directed intent are critical but difficult to capture through static visual frames alone.

**LLM Limitations and VLM Advantages**  Figs. 7a and 7b present two cases from the MetaWorld *Door Open* task where VLMs outperform LLMs in preference evaluation. Fig. 7a highlights a subtle but critical spatial

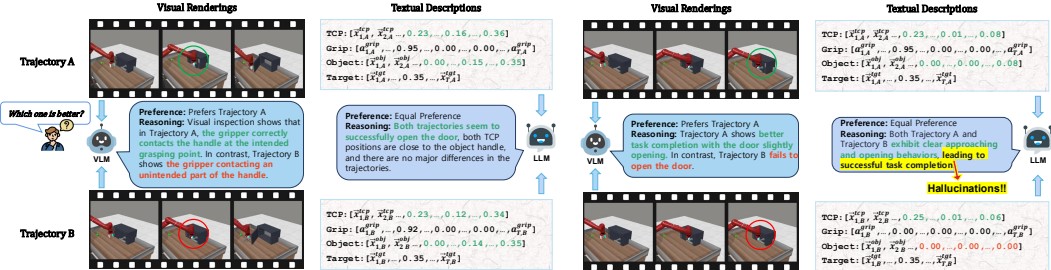

(a) VLM correctly identifies a grasp error in Trajectory B, where the gripper contacts the hinge side instead of the intended handle region, while the LLM fails to detect this misalignment and assigns equal preference.

(b) VLM accurately recognizes that only Trajectory A successfully opens the door, while the LLM incorrectly infers that both trajectories are equally successful.

Figure 7: Examples where VLM outperforms LLM in preference evaluation, highlighting the advantages of direct visual perception for detecting spatial errors, physical contact quality, and task success.

distinction. In Trajectory A, the gripper correctly makes contact with the handle at its intended grasp region, leading to a successful interaction. In contrast, Trajectory B misaligns the gripper with the hinge side of the door, resulting in a mechanically incorrect grasp. This error is visible in the red-circled region of Trajectory B. The VLM, grounded in direct visual perception, correctly identifies this spatial discrepancy and selects Trajectory A as the preferred option. However, the LLM, which relies on abstract state variables such as TCP position, gripper state, and object coordinates, fails to capture this fine-grained spatial error, instead outputting equal preference for both trajectories. This occurs because the LLM lacks direct perceptual grounding, making it blind to critical spatial misalignments that are immediately evident in visual frames.

Fig. 7b presents a more pronounced failure case. Here, only Trajectory A successfully opens the door, as indicated by the green circle, while Trajectory B fails to produce any meaningful outcome, as marked by the red circle. Despite this clear visual difference, the LLM mistakenly infers that both trajectories are equally successful, likely due to similar coordinate sequences that superficially resemble goal-directed behavior. This is a classic hallucination error, where the LLM abstracts away from the actual physical outcome, ignoring critical perceptual cues. In contrast, the VLM, which directly observes the task's visual consequences, correctly identifies Trajectory A as superior.

Together, these examples highlight the limitations of relying solely on language-based state representations for preference evaluation. Without direct visual input, LLMs can miss critical spatial alignments, physical contacts, and fine-grained task completions, leading to misleading or incorrect preferences. In contrast, VLMs are inherently suited to capture these spatial relationships, making them indispensable for evaluating precise robotic interactions. This complementarity underscores the need for a multimodal preference model that integrates the perceptual grounding of VLMs with the temporal and semantic insights of LLMs, enabling more robust and context-aware trajectory evaluation.

## A.2  Quantitative Analysis of Trajectory Context Influence on Each Modality

From the qualitative examples above, it is clear that each feedback modality has its own potential limitations yet complementary strengths. We can observe that the reliability of these labels can vary significantly depending on the specific *trajectory context*, such as whether the trajectories contain visually distinguishable features or exhibit meaningful temporal progression.

To fully leverage these complementary capabilities, it is essential to quantitatively analyze how trajectory content impacts the labeling reliability of each feedback modality. Given our observations that VLMs tend to excel at analyzing spatial cues from visual renderings, we hypothesize that VLM labels should be more accurate when the visual discriminability (i.e., spatial differences) between trajectories is higher. Similarly, we expect LLM labels to be more accurate when the temporal discriminability (i.e., time-dependent behavioral differences) between trajectories is more pronounced.

To validate these hypotheses, we conducted a quantitative analysis to assess the impact of trajectory context on labeling accuracy for each modality. Specifically, we examined the correlation between the degree of context contrast, i.e., VDHigh defined in Eq. 9 for VLMs and TDHigh defined in Eq. 10 for LLMs, and the corresponding labeling accuracy.

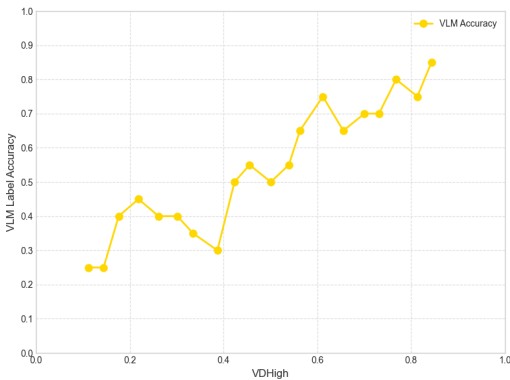 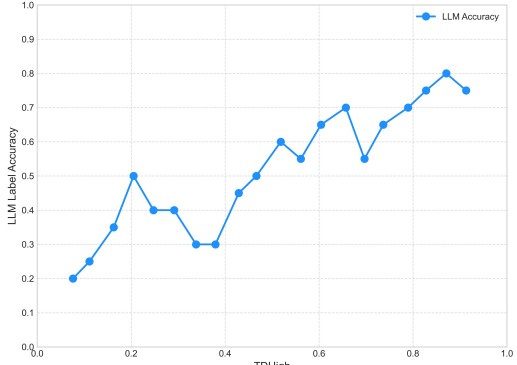

(a) VLM label accuracy generally increases as `VDHigh` increases.

(b) LLM label accuracy generally increases as `TDHigh` increases.

Figure 8: Impact of trajectory context on labeling accuracy of each feedback modality.

For this analysis, we sampled 200 trajectory pairs from the Button-press task in the MetaWorld benchmark and clustered them into 20 groups separately based on their `VDHigh` and `TDHigh` values, respectively. We then collected human expert labels as ground truth and computed the labeling accuracy of both VLMs and LLMs under varying levels of context contrast. The results are presented in Fig. 8a and Fig. 8b.

As shown in Fig. 8a, VLM label accuracy generally increases with higher `VDHigh`, confirming that VLMs benefit from stronger visual cues when assigning preference labels. Similarly, Fig. 8b demonstrates that LLM label accuracy improves as `TDHigh` increases, supporting our hypothesis that LLMs are more effective at capturing temporal dependencies in trajectory comparisons.

These results validate the conflict resolution rules in our PSL-based inter-modal fusion, confirming that context-specific predicates (`VDHigh` and `TDHigh`) can enhance preference labeling reliability by aligning each modality with its respective strengths, in addition to considering modality-specific confidence. This structured approach not only reduces noise but also improves the overall robustness and interpretability of the inter-modal preference aggregation process.

## B  Details of Keyframe Extraction

In this section, we provide details of our hybrid method in extracting keyframes from robot trajectories. Given a trajectory $\sigma$, we propose three complementary methods for extracting keyframes. The final keyframe set, denoted as $kvis(\sigma)$, is formed by taking the union of the selected frames from all three methods along with the first and last frames of the trajectory. Below, we detail each method:

### B.1  Near-Zero Velocity Detection

Near-zero velocity detection aims to identify moments where the agent's motion is minimal, which often corresponds to sub-goal completions (e.g., grasping, placing) or phase boundaries in locomotion tasks. This method leverages the intuition that significant actions or transitions in a task are often preceded or followed by periods of reduced movement or even complete stillness. In our setting, a robot trajectory is defined as a sequence of state-action pairs:

$$\sigma = \{(s_t, a_t)\}_{t=1}^T, \tag{17}$$

where each time step $t$ consists of a state vector $s_t \in \mathbb{R}^{d_s}$ and an action vector $a_t \in \mathbb{R}^{d_a}$. The state vector $s_t$ typically encapsulates the environment and robot configuration at time $t$, such as target position, joint angles, end-effector pose (position and orientation), and potentially other relevant sensor readings. The action vector $a_t$ represents the commands issued to the robot's actuators, like joint velocities or torques. The combined observation-action vector is then given by:

$$\mathbf{x}_t = [s_t, a_t] \in \mathbb{R}^{d_s + d_a}. \tag{18}$$

By concatenating the state and action vectors, we create a comprehensive representation of the robot's instantaneous context, capturing both its current configuration and the commands being executed. The L2 velocity between consecutive frames is defined as:

$$v_t = \|\mathbf{x}_{t+1} - \mathbf{x}_t\|_2 = \sqrt{\sum_{i=1}^{d_s+d_a} (x_{t+1,i} - x_{t,i})^2} \tag{19}$$

where $\mathbf{x}_t$ captures both the state and action at each time step. A timestep $t+1$ is selected as a keyframe if the velocity magnitude is below a predefined threshold:

$$\mathcal{K}_{\text{zero}} = \{t+1 \mid v_t < \delta_v\} \tag{20}$$

where $\delta_v$ is the velocity threshold that controls the sensitivity to small movements. A smaller value of $\delta_v$ results in the selection of more keyframes, capturing even subtle pauses in the motion. Conversely, a larger $\delta_v$ only identifies keyframes where the robot comes to a more significant halt.

We set $\delta_v$ to 0.005 for manipulation tasks in MetaWorld, 0.025 for high-dimensional manipulation tasks in ManiSkill2, and 0.065 for locomotion tasks in DeepMind Control Suite (DMC). These values are chosen based on task characteristics and observation-action dimensionality across environments. MetaWorld tasks typically involve smooth, low-dimensional end-effector control, where even small pauses (e.g., before grasping) are meaningful and should be captured. ManiSkill2 features more complex manipulation scenarios with higher-dimensional state-action spaces. Therefore, a moderately larger threshold is needed to account for naturally higher baseline velocities. In contrast, DMC locomotion tasks involve rhythmic, continuous movement patterns and higher physical velocity variance, requiring a larger threshold to isolate meaningful slowdowns (e.g., stance phase transitions or contact events).

## B.2 Smoothing Residual Peaks

This method aims to identify keyframes that correspond to significant changes in the robot's motion, such as sharp turns, sudden accelerations or decelerations, and other high-curvature segments of the trajectory. By comparing the original, raw trajectory with a smoothed version, we can highlight these abrupt transitions as deviations with large residual errors. The residual error is computed as:

$$e_t = \|\mathbf{x}_t - \tilde{\mathbf{x}}_t\|_2 = \sqrt{\sum_{i=1}^{d_s+d_a} (x_{t,i} - \tilde{x}_{t,i})^2} \tag{21}$$

where $\tilde{\mathbf{x}}_t$ is the smoothed trajectory point, typically computed using a moving average filter:

$$\tilde{\mathbf{x}}_t = \frac{1}{2k+1} \sum_{j=-k}^{k} \mathbf{x}_{t+j} \tag{22}$$

Here, the moving average filter acts as a low-pass filter, effectively smoothing out high-frequency components in the trajectory that correspond to rapid changes in motion. The smoothing window size is controlled by the parameter $k$. A larger $k$ produces a smoother baseline trajectory $\tilde{\mathbf{x}}_t$, increasing sensitivity to local spikes in the raw trajectory $\mathbf{x}_t$. The value of $k$ should be chosen based on the typical temporal scale of meaningful motion transitions in the task: short for fine-grained manipulation, longer for rhythmic locomotion.

Frames with the top $K$ largest residual errors are selected:

$$\mathcal{K}_{\text{smooth}} = \text{Top-}K(\{e_t \mid e_t > \delta_e\}) \tag{23}$$

where $K$ is the maximum number of keyframes to extract using this method, and $\delta_e$ is a residual error threshold that filters out minor fluctuations due to noise.

We set $k = 2$, $K = 5$, and $\delta_e = 0.01$ for MetaWorld, reflecting short-horizon, low-frequency transitions typical of single-object manipulation. For ManiSkill2, which features higher-dimensional and more dynamic manipulation behaviors, we use a slightly larger window $k = 4$, $K = 8$, and a relaxed threshold $\delta_e = 0.02$ to tolerate high-frequency noise. For DMC locomotion tasks, we set $k = 6$, $K = 10$, and $\delta_e = 0.04$, as abrupt transitions (e.g., foot contact, turning) occur at lower frequency but with higher magnitude. These settings ensure that the residual-based method captures meaningful structure transitions across environments with diverse temporal dynamics.

## B.3 Change Point Detection (CPD)

Change Point Detection (CPD) offers a powerful approach to segment a robot's trajectory into distinct phases characterized by different motion dynamics. These phases often correspond to meaningful parts of the task, such as reaching, grasping, moving, and releasing an object. By identifying the boundaries between these phases, we can extract keyframes that represent transitions between different stages of the robot's behavior.

We implement the Pruned Exact Linear Time (PELT) algorithm using the `ruptures` library[2], which minimizes a penalized cost function to efficiently detect change points in the trajectory data:

$$\min_{1 < \tau_1 < \ldots < \tau_M < T} \left[ \sum_{m=0}^{M} \mathcal{C}(\sigma_{\tau_m : \tau_{m+1}}) + \beta M \right] \tag{24}$$

Here, $\tau_0 = 1$ and $\tau_{M+1} = T$ represent the start and end of the trajectory, and $\tau_1, \ldots, \tau_M$ are the detected change points that segment the trajectory into $M + 1$ homogeneous phases. The cost function $\mathcal{C}(\cdot)$ quantifies intra-segment consistency, typically using the sum of squared L2 deviations. The penalty term $\beta$ controls the trade-off between segmentation fidelity and the number of segments: larger $\beta$ values lead to coarser segmentations (fewer change points), while smaller values allow for finer-grained segmentations.

We set $\beta = 20$ for MetaWorld, which generally contains short-horizon manipulation behaviors with 3–5 semantically distinct phases. For high-dimensional manipulation in ManiSkill2, we use $\beta = 30$, reflecting its more complex task structures with frequent mid-task corrections and multiple contact events. For locomotion tasks in DMC, we set $\beta = 40$, as rhythmic gaits tend to repeat smoothly over time, and fewer change points are expected to capture major phase transitions (e.g., stance-to-swing). These values are chosen to reflect the temporal structure and motion complexity of each task environment. The resulting change points $\tau_M$ are used directly as keyframes:

$$\mathcal{K}_{\text{cpd}} = \{\tau_1, \tau_2, \ldots, \tau_M\} \tag{25}$$

These change points correspond to moments where the motion dynamics of the trajectory undergo a significant shift, making them valuable for summarizing high-level behavioral transitions during task execution.

## B.4 Combining Methods for Keyframe Extraction

To achieve a more comprehensive and robust selection of keyframes, we recognize that each of the aforementioned methods captures different aspects of salient motion. Near-zero velocity detection identifies periods of stagnation, smoothing residual peaks highlights abrupt changes, and change point detection pinpoints transitions between distinct motion phases. By combining the keyframes identified by these complementary approaches, we aim to obtain a more complete representation of the robot's task execution. The final set of visually significant keyframes, denoted as $\mathcal{K}_{\text{vis}}$, is obtained by taking the union of the keyframe sets generated by each individual method ($\mathcal{K}_{\text{zero}}$, $\mathcal{K}_{\text{smooth}}$, and $\mathcal{K}_{\text{cpd}}$). Additionally, to ensure that the beginning and the end of the entire trajectory are always included in our set of keyframes, we explicitly add the first frame (index 1) and the last frame (index $T$) to the combined set:

$$\mathcal{K}_{\text{vis}} = \mathcal{K}_{\text{zero}} \cup \mathcal{K}_{\text{smooth}} \cup \mathcal{K}_{\text{cpd}} \cup \{1, T\} \tag{26}$$

Empirically, the parameter settings specified above yield about 10-20 keyframes per trajectory on MetaWorld, 15–20 on ManiSkill2, and on DMC, providing a balanced summary without overwhelming the downstream multimodal evaluator.

## C   Prompts and Example Outputs

### C.1   Synthetic Preference Generation

In this section, we provide prompt templates used to elicit the preference judgments from the VLM and LLM in Fig. 9. We also provide the details of the {task_description}, which at run-time is filled with the natural-language goal statement for the current environment, in Table 1.

### C.2   Foresight Trajectory Generation

In this section, we provide the prompt template for foresight trajectory generation, Fig. 10a, as well as example inputs of the prompt with MetaWorld's DoorOpen task, as shown in Fig. 10b. The foundation model is instructed first to sketch a high-level motion plan, then to translate that plan into executable Python code, and finally to diversify the result by changing initial conditions and motion parameters. An example of the resulting executable Python code for the Door Open task, generated using the prompt, is provided in Fig. 11.

---

[2]https://github.com/deepcharles/ruptures

Table 1: Task description for each task used in our experiments.

| Task Name | Task Description |
|---|---|
| *Button Press* | to press a button on a surface |
| *Door Open* | to open a door with a revolving joint. |
| *Sweep Into* | to sweep the object into the target area |
| *PickSingleYCB* | to pick up a random object sampled from the YCB dataset and move it to a random goal position |
| *StackCube* | to pick up a red cube and stack it on top of a green cube and let go of the cube without it falling |
| *PegInsertionSide* | to pick up an orange-white peg and insert the orange end into the box with a hole in it |
| *Hopper Stand* | to stabilize a planar one-legged hopper initialized in a random pose, encouraging upright posture with minimal torso height loss |
| *Walker Walk* | to control a planar bipedal walker to move forward with a target velocity. |

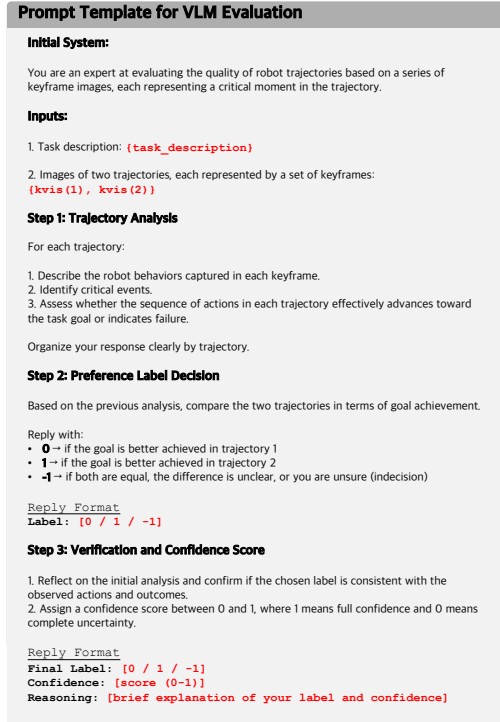

(a) Illustration of the VLM prompt.

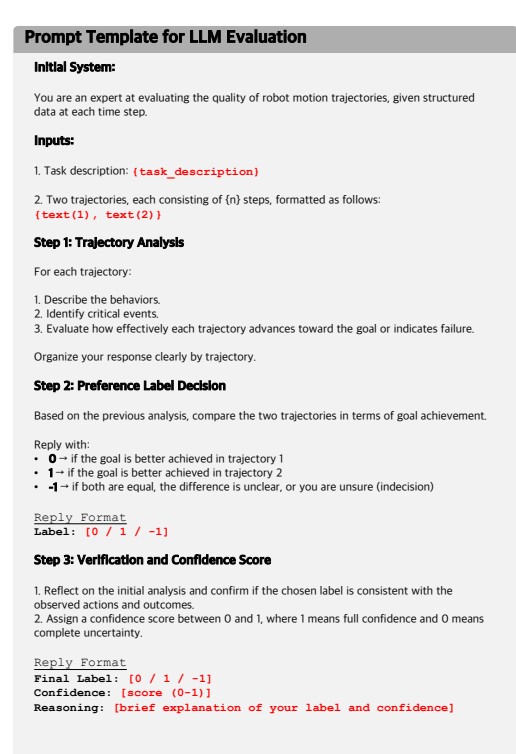

(b) Illustration of the LLM prompt.

Figure 9: Prompt templates used for synthetic preference generation in PRIMT.

## C.3   Hindsight Trajectory Augmentation

The prompt template for counterfactual trajectory augmentation is shown in Fig. 12. In this template, `kvist1`, `kvist2` denote the keyframe indices extracted using the methods described in SectionB. Examples of the generated counterfactuals are provided in Fig. 20.

# D   PSL Inference Details

We provide additional details on the Probabilistic Soft Logic (PSL) inference procedure used in our inter-modal preference fusion module.

**Initial System:**

You are a robotics planning agent tasked with generating diverse, semantically meaningful robot trajectories for simulated manipulation or locomotion tasks. Your goal is to create high-quality, task-aligned trajectories that follow physical constraints and can be used for boostrapping training.

**Inputs:**

1. Task description: **{task_description}**
2. Trajectory Format and Constraints:
**{Trajectory_Format}{Trajectory_Constraints}**
3. Environment Setup: **{Environment_Setup}**

**Step 1: High-Level Motion Plan Decomposition**

Decompose the task into 3-5 discrete high-level phases. For each phase, clearly specify:

- Phase Name
- Subgoal: What condition or spatial configuration defines success at this phase?
- Motion Strategy: What type of movement or action is needed?
- Constraint Handling: How to account for physical limits, contact, kinematic constraints, etc.

Reply Format
```
Phase Plan = [
  {"phase": ..., "subgoal": ..., "strategy": ...,
"constraints": ...},
  ...]
```

**Step 2: Code Translation for Motion Primitives**

- Translate the high-level action plan into executable Python code.
- Implement each phase using structured, reusable motion primitives that can adapt to different task contexts.
- Key considerations:
  • Control Logic: Implement continuous control logic for smooth transitions.
  • Parameter Flexibility: Allow inputs of different initial conditions and motion strategy parameters.

Reply Format
**{code_return_format}**

**Step 3: Trajectory Diversification and Sampling**

Generate **{n}** diverse trajectory samples by varying both:

- Initial Conditions: Sample from a distribution of start states (e.g., different object locations or robot poses)
- Strategy Parameters: Adjust trajectory-level control parameters (e.g., approach angle, lift height, motion speed)

Run the code multiple times to populate the trajectory buffer with diverse samples.

Reply Format
```
[{
    "traj_id": "fore_001",
    "init_state": {...},
    "strategy_config": {...},
    "trajectory": [...]
  },...]
```

**Example Inputs and Requirements on MetaWorld**

**Trajectory_Format:**

```
1. [0:2]   - End-effector (TCP) position in Cartesian space,
normalized ∈ [0, 1]
2. [3]     - Gripper opening level: 0 = closed, 1 = open
3. [4:6]   - Object position (door handle), 3D, normalized ∈ [0,
1]
4. [7:10]  - Object orientation (quarternion, 4D)
5. [11:13] - TCP delta action (Δx, Δy, Δz)
6. [14]    - Gripper torque control (normalized)
```

**Trajectory_Constraints:**

```
- Trajectory must contain exactly 100 steps

- Each transition must be smooth (no large discontinuities between
consecutive steps)

- Avoid collisions with the table (assumed z > 0.05)

- Action values [11:14] (TCP deltas and gripper torque) must be
consistent with state changes in [0:3] and [3]
```

**Environment_Setup:**

```
- Robot: Sawyer arm with a top-down gripper

- Workspace frame:
• x-axis: left → right
• y-axis: near → far
• z-axis: bottom → top

- Gripper logic:
• Open (1.0) during approach
• Close (0.0) during interaction
• Open again during release

- Initial values provided:
• obj_pos_start: Initial object position (3D): [0.1. 0.95, 0.15]
• obj_ori_start: Initial object orientation (4D): [0, 0, 0, 0]
• target_pos: Goal handle position (3D): [-0.2, 0.2, 0.15]
```

**Code_Return_Format:**

```
def generate_trajectory(init_state, strategy_config, env_params):
    """
    Generate a full trajectory by sequentially composing
functional motion phases.
    """

    trajectory = []

    # === Phase 1 ===
    traj_phase1 = phase_approach(...)
    trajectory += traj_phase1

    # === Phase N ===
    traj_phase_n = phase_release(...)
    trajectory += traj_phase_n

    return trajectory
```

(a) Foresight trajectory generation prompt.    (b) Example inputs on the Metaworld environment.

Figure 10: Prompt templates for foresight trajectory generation in PRIMT.

**Variables.** Let $X$ denote the set of observed atoms, which include modality-specific predictions and trajectory-level context features, such as $\texttt{VLMLabel}(\Upsilon)$, $\texttt{LLMLabel}(\Upsilon)$, $\texttt{ConfHigh}(M)$, $\texttt{VDHigh}$, and $\texttt{TDHigh}$. Let $Y$ denote the set of target atoms, corresponding to the final preference decision to be inferred: $\texttt{FinalLabel}(\Upsilon)$ for $\Upsilon \in \{-1, 0, 1\}$.

**HL-MRF Formulation.** PSL defines a hinge-loss Markov random field (HL-MRF) over the target atoms $Y$, representing a log-linear probabilistic model:

$$P(Y \mid X) = \frac{1}{Z} \exp\left(-\sum_{i=1}^{m} w_i \cdot \phi_i(Y, X)\right), \tag{27}$$

where $w_i$ is the weight assigned to the $i$-th rule, and $\phi_i(Y, X) = [\max(0, \ell_i(Y, X))]^p$ is the relaxed hinge-loss potential derived from the rule's linear distance to satisfaction $\ell_i$. The normalization constant $Z$ integrates over the feasible soft assignment space:

$$Z = \int_{Y \in [0,1]^n} \exp\left(-\sum_{i=1}^{m} w_i \cdot \phi_i(Y, X)\right) dY. \tag{28}$$

Inference is performed via convex optimization over continuous variables $Y$, subject to any additional linear constraints.

**Łukasiewicz Relaxation.** Given two grounded atoms $A_1, A_2 \in [0, 1]$, PSL uses the following relaxation rules for conjunction, disjunction, and negation:

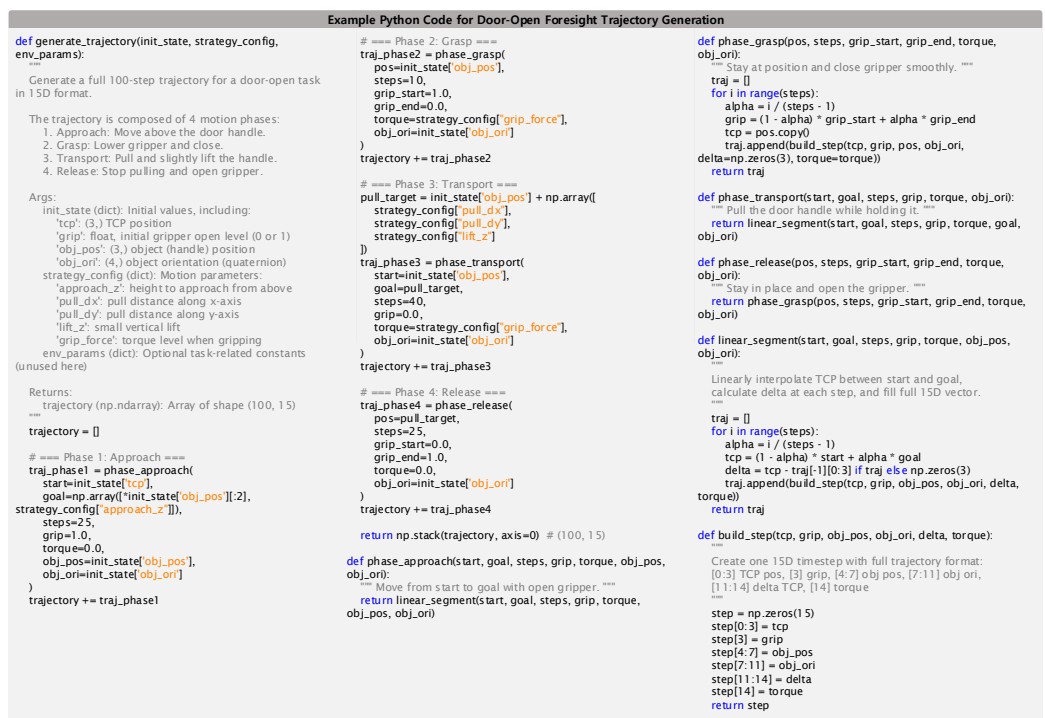

Figure 11: An example of the resulting executable Python code for the Door Open task.

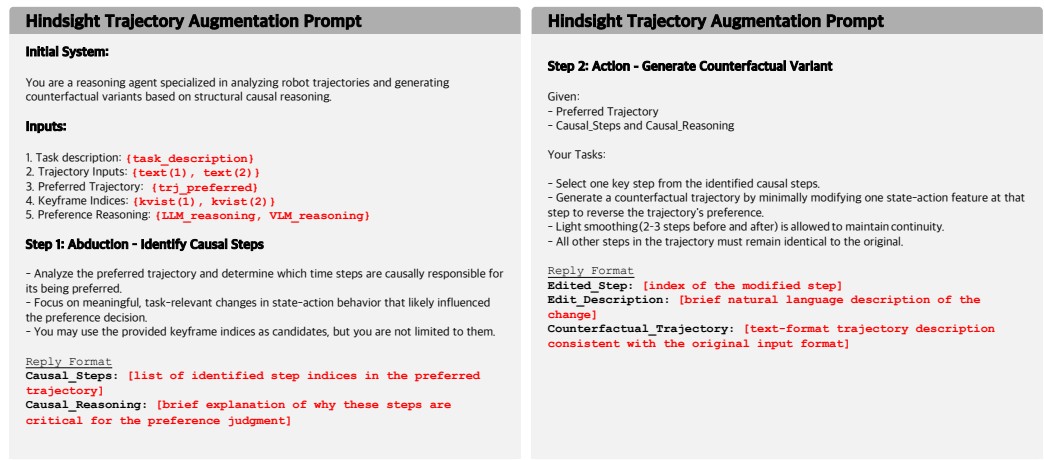

Figure 12: Prompt template used for counterfactual trajectory augmentation in PRIMT

$$A_1 \widetilde{\wedge} A_2 = \max\{0, A_1 + A_2 - 1\} \tag{29}$$

$$A_1 \widetilde{\vee} A_2 = \min\{A_1 + A_2, 1\} \tag{30}$$

$$\widetilde{\neg} A_1 = 1 - A_1 \tag{31}$$

These allow soft rules to be represented as continuous linear functions over truth values, enabling efficient convex inference.

**Template Rule Grounding.** PSL rules are first written as templates over logical variables (e.g., labels $\Upsilon$ or modalities $M$), and then instantiated into multiple ground rules. For example, the template:

$$\forall \Upsilon, M : \texttt{IsAgree}(\Upsilon) \wedge \texttt{ConfHigh}(M) \rightarrow \texttt{FinalLabel}(\Upsilon)$$

is expanded into six rules by enumerating all $\Upsilon \in \{-1, 0, 1\}$ and $M \in \{\texttt{VLM}, \texttt{LLM}\}$, such as:

$$\texttt{IsAgree}(1) \wedge \texttt{ConfHigh}(\texttt{LLM}) \rightarrow \texttt{FinalLabel}(1)$$

**Example Rule Expansion.** Consider the rule:

$$\texttt{VLMLabel}(1) \wedge \texttt{ConfHigh}(\texttt{VLM}) \wedge \texttt{VDHigh} \rightarrow \texttt{FinalLabel}(1)$$

Its logical equivalent is:

$$\neg\texttt{VLMLabel}(1) \vee \neg\texttt{ConfHigh}(\texttt{VLM}) \vee \neg\texttt{VDHigh} \vee \texttt{FinalLabel}(1)$$

Using Łukasiewicz semantics, the corresponding relaxed satisfaction distance is:

$$\ell = \texttt{VLMLabel}(1) + \texttt{ConfHigh}(\texttt{VLM}) + \texttt{VDHigh} - \texttt{FinalLabel}(1) - 3$$

and the resulting hinge-loss potential is:

$$\phi = [\max(0, \ell)]^p = [\max(0, \texttt{VLMLabel}(1) + \texttt{ConfHigh}(\texttt{VLM}) + \texttt{VDHigh} - \texttt{FinalLabel}(1) - 3)]^p$$

# E Details on Task Environments

In this section, we provide details on the tasks used in our experiments.

## E.1 MetaWorld

**Button Press** The Button Press task requires the robot to manipulate its end-effector to press a specific button located on a surface. The goal is to make contact with and depress the button. The reward is typically sparse, given only when the button is successfully pressed. This task tests the robot's ability to achieve precise movements and interact with small objects in the environment, as visualized in **Fig. 13a**.

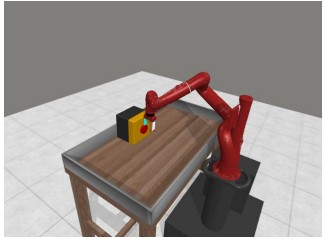 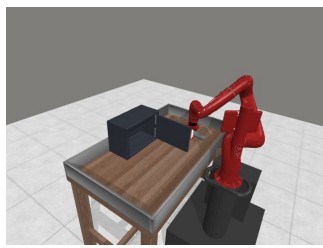 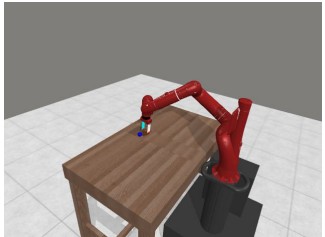

| (a) Button Press Task | (b) Door Open Task | (c) Sweep Into Task |

Figure 13: Visualizations of MetaWorld Tasks

**Door Open** The Door Open task involves the robot grasping the handle of a hinged door and opening it to a desired angle. The robot must first reach and grasp the handle, then apply the appropriate force and motion to swing the door open. This task assesses the robot's ability to perform sequential manipulation actions and interact with articulated objects, as shown in **Fig. 13b**.

**Sweep Into** The Sweep Into task challenges the robot to use its end-effector (or an object held by it) to sweep a target object into a designated goal region. This requires the robot to make contact with the object and apply a sweeping motion to push it into the target. This task evaluates the robot's ability to perform planar manipulation and reason about pushing dynamics, illustrated in **Fig. 13c**.

## E.2 ManiSkill

**PickSingleYCB** The PickSingleYCB task involves the robot picking up a single object from the YCB object set. The robot must perceive the object, plan a grasp, execute the grasp, and lift the object to a desired height or location. This is a fundamental object manipulation task that tests grasping and lifting skills, as visualized in **Fig. 14a**.

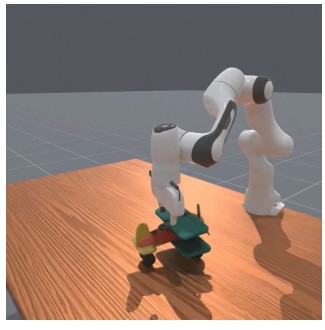
(a) PickSingleYCB Task

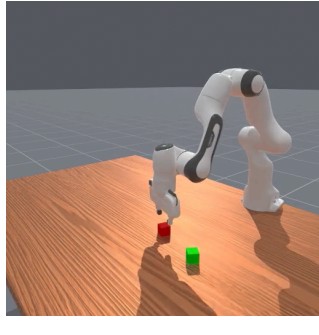
(b) StackCube Task

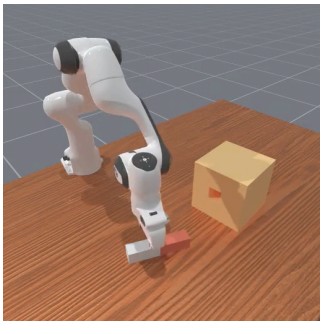
(c) PegInsertionSide Task

Figure 14: Visualizations of ManiSkill Tasks

**StackCube**  The StackCube task requires the robot to pick up one or more cubes and stack them on top of each other in a stable configuration. This task builds upon basic picking and placing skills and introduces the challenge of achieving a stable multi-object arrangement, as shown in **Fig. 14b**.

**PegInsertionSide**  The PegInsertionSide task involves the robot inserting a peg into a hole on the side of a surface. This task requires precise alignment and control of the robot's end-effector to successfully insert the peg without collision. It tests fine manipulation and spatial reasoning, illustrated in **Fig. 14c**.

### E.3  DeepMind Control Suite

**Hopper Stand**  The Hopper Stand task from the DeepMind Control (DMC) suite involves a single-legged hopping robot. The goal is to control the robot to stand upright and maintain its balance without falling. This task assesses the agent's ability to learn stable control policies for a dynamically challenging system, as visualized in **Fig. 15a**.

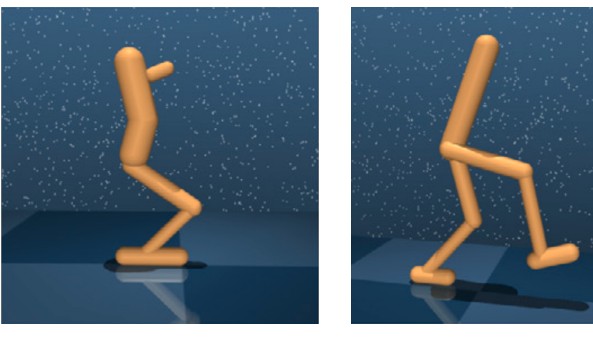
(a) Hopper Stand Task           (b) Walker Walk Task

Figure 15: Visualizations of DeepMind Control Suite Tasks

**Walker Walk**  The Walker Walk task from the DeepMind Control (DMC) suite features a bipedal walking robot. The objective is to control the robot to walk forward with a consistent velocity without falling. This task evaluates the agent's ability to learn complex locomotion gaits and maintain stability in a more articulated system, as shown in **Fig. 15b**.

## F  Additional Implementation Details

In this section, we provide further implementation details of our experiments.

### F.1  Baselines

We implemented the baselines RL-VLM-F [14], RL-SaLLM-F [16], and PrefCLM [15] using the source code released by the authors. To eliminate non-model differences, we replaced the original prompts in RL-VLM-F and RL-SaLLM-F with the same three-step chain-of-thought prompt used in PRIMT. This three-step prompt can

be considered an enhanced version of the two-step reasoning used in their original papers, with an additional self-verification step. The trajectory inputs are also the same as in PRIMT: $kvis(\sigma)$ for RL-VLM-F and $text(\sigma)$ for RL-SaLLM-F.

The only exception is PrefCLM. This method assumes access to the environment code, which it uses to generate diverse evaluation functions by prompting multiple LLMs, and then fuses the resulting preference scores using Dempster–Shafer theory. Due to its reliance on environment-specific code, we retain its original setup. For fairness, we evaluate PrefCLM under its zero-shot variant, which does not involve few-shot expert selection of the generated functions and human-in-the-loop correction. The crowd size is set to 10, following the original configuration in the paper.

We implemented the PrefMul baseline ourselves by feeding the multimodal trajectory representations, i.e., $text(\sigma)$ and $kvis(\sigma)$, into a multimodal LLM (e.g., `gpt-4o`), using the same CoT prompt as PRIMT to elicit the preference label.

The PrefGT baseline follows the scripted teacher approach introduced in [8]. It serves as an upper-bound oracle that has complete access to the benchmark's ground-truth reward function. Although such access is infeasible in real-world robotic systems, it provides a useful reference to assess the best possible performance that any preference-based learning method could achieve.

For any pair of trajectories $\sigma^A$ and $\sigma^B$, the oracle computes their cumulative reward returns:

$$R(\sigma) = \sum_{t=1}^{T} r(s_t, a_t)$$

where $r(s_t, a_t)$ is the environment's ground-truth reward at time step $t$. The oracle then assigns the preference label as:

$$\Upsilon = \begin{cases} 1 & \text{if } R(\sigma^A) > R(\sigma^B) \\ 0 & \text{if } R(\sigma^A) < R(\sigma^B) \\ -1 & \text{if } R(\sigma^A) = R(\sigma^B) \end{cases}$$

## F.2  Reward Learning

We adopt PEBBLE [7] as the PbRL backbone for all methods. PEBBLE first performs unsupervised pre-training to maximize state entropy and initialize the policy, followed by off-policy reinforcement learning using Soft Actor-Critic (SAC) for policy optimization. During training, all reward values in the replay buffer are relabeled whenever a new reward model is learned.

For the reward model, we implement the image-based reward network as described in their original paper for RL-VLM-F. For all other baselines using the standard PEBBLE [7] reward model, we adopt a 3-layer ensemble architecture following their design.

For selecting informative queries, we adopt an uncertainty-based query selection strategy following [8]. Specifically, we measure the uncertainty of preference predictions either by computing the variance across an ensemble of preference predictors. We then select the top-$N_{\text{query}}$ trajectory segment pairs with the highest uncertainty as query candidates to elicit preference labels.

For all experiments conducted in this work, the hyperparameter settings used for reward learning are summarized in Table 2.

Table 2: Hyperparameters used for reward learning.

| Hyperparameter | Value |
| --- | --- |
| Trajectory segment length | 100 |
| Feedback frequency | 5000 |
| Maximum feedback samples | 20000 |
| Number of foresight trajectories | 200 |
| Max counterfactuals per trajectory | 5 |

## F.3  Policy Learning

For all methods evaluated in this work, we follow PEBBLE [7] and adopt Soft Actor-Critic (SAC) as the off-policy reinforcement learning algorithm. To ensure fair comparison, all methods share the same actor-critic architecture and hyperparameter settings during policy learning.

Throughout training, we use the same network configurations and learning schedules as those in the original PEBBLE implementation. A summary of these hyperparameters is provided in Table 3.

All experiments were conducted on a workstation equipped with five NVIDIA RTX 4090 GPUs.

Table 3: Hyperparameters used for SAC.

| Hyperparameter | Value | Hyperparameter | Value |
|---|---|---|---|
| Initial temperature | 0.1 | Batch Size | 1024 |
| Learning rate | 0.0003 | Optimizer | Adam |
| Critic target update freq | 2 | Critic EMA $\tau$ | 0.005 |
| $(\beta_1, \beta_2)$ | (0.9, 0.999) | Discount $\gamma$ | 0.99 |
| Hidden units per each layer | 1024 | | |

# G   Additional Experimental Results

In this section, we provide additional experimental results to further support our main findings. These include:

- More visualizations of label distributions and learned reward outputs across tasks.
- An ablation study on the impact of foundation model (FM) backbone selection.
- Policy visualizations comparing PRIMT and baseline behaviors.
- Qualitative analysis of both the foresight and hindsight trajectory generation modules.
- Real-world deployment on a Kinova Jaco robot.

## G.1   More Visualizations of Label Distributions and Learned Reward Outputs

We present additional visualizations of label distributions and learned reward outputs on other tasks in Figs. 16 and 17, in addition to those shown in Fig. 4.

We observe that PRIMT consistently produces higher-quality synthetic feedback with fewer indecisive labels. Furthermore, the reward functions learned by PRIMT exhibit more precise state-action-level credit assignment, resulting in reward patterns that better align with ground-truth task progress.

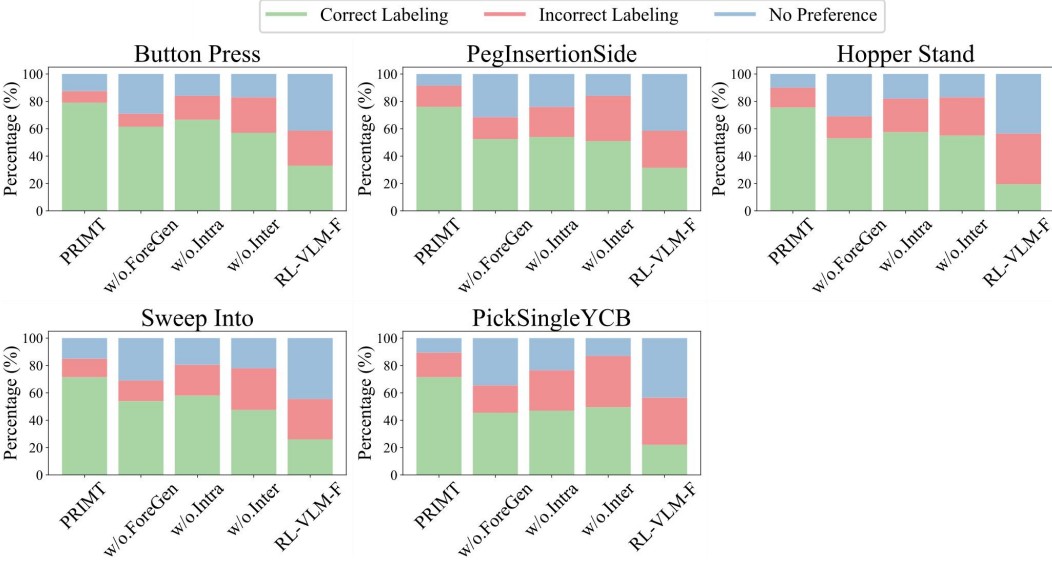

Figure 16: Extra visualizations of the distribution of preference labels, showing the proportion of correct, incorrect, and indecisive labels across different methods.

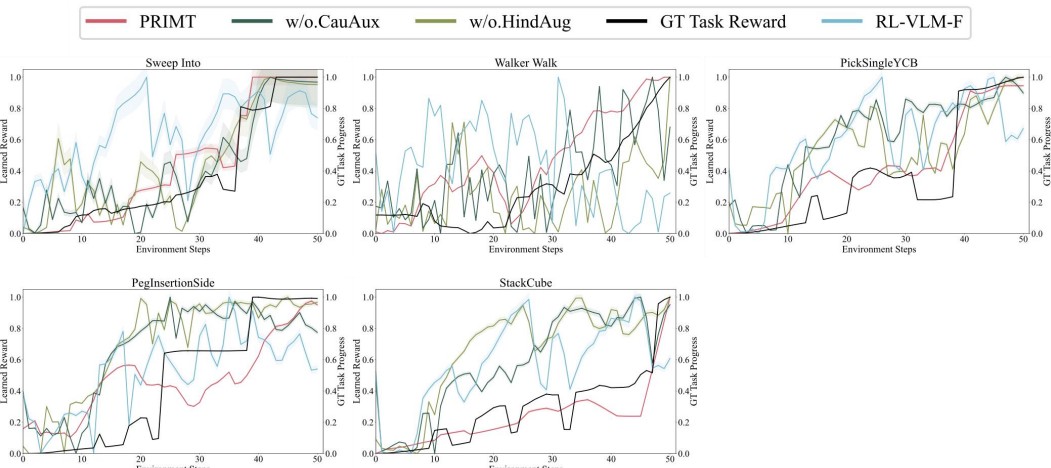

Figure 17: Extra visualizations of the reward alignment, comparing the learned reward outputs of PRIMT, ablations, and baselines against ground-truth reward.

## G.2 Ablation Study on FM Backbone Selection

We further conduct an ablation study to investigate the influence of the FM backbone on two representative tasks: *Door Open* from the MetaWorld benchmark and *PegInsertionSide* from the ManiSkill benchmark. We report the performance of PRIMT and the best-performing baseline on each task using `gpt-4o`, and compare it to the same setup using `gpt-4o-mini`, a weaker model variant.

As shown in Fig. 18, both methods experience a performance drop when using `gpt-4o-mini`, which is expected due to the reduced reasoning and perception capabilities of the smaller model. However, PRIMT demonstrates greater robustness, maintaining performance more effectively compared to the best-performing baseline. This again highlights the benefit of our hierarchical fusion and trajectory synthesis design in improving generalizability under lower-capacity foundation models.

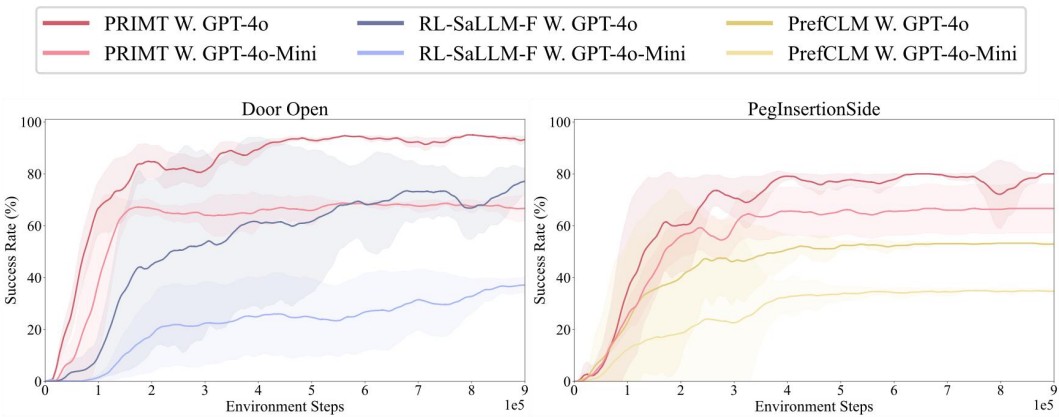

Figure 18: Ablation study on FM backbone selection.

## G.3 Qualitative Analysis of the Foresight Trajectory Generation Module

To better understand the behaviors produced by our foresight trajectory generation module, we visualize several examples of LLM-generated trajectories and compare them to early-stage random explorations. The visualizations are shown in Fig. 19.

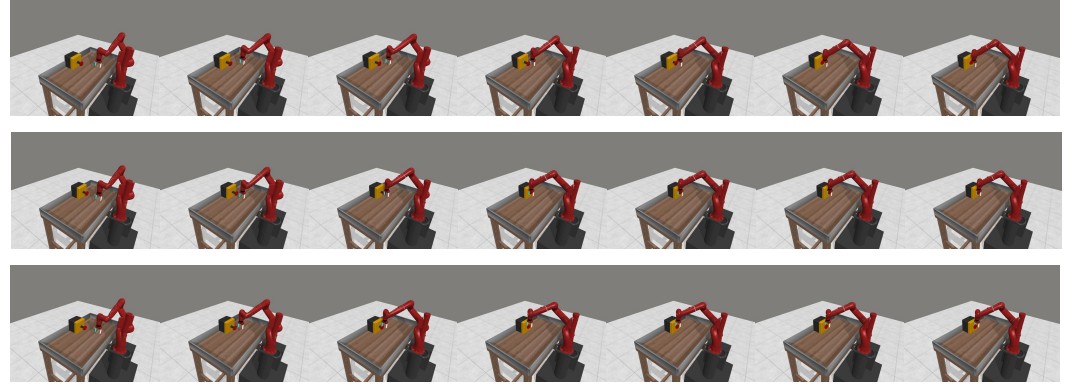

(a) LLM-generated trajectories that successfully complete the task, showing diversity in robot starting positions (e.g., the first row vs. the other two), and in strategies such as varying gripper height to press the button (second and third rows).

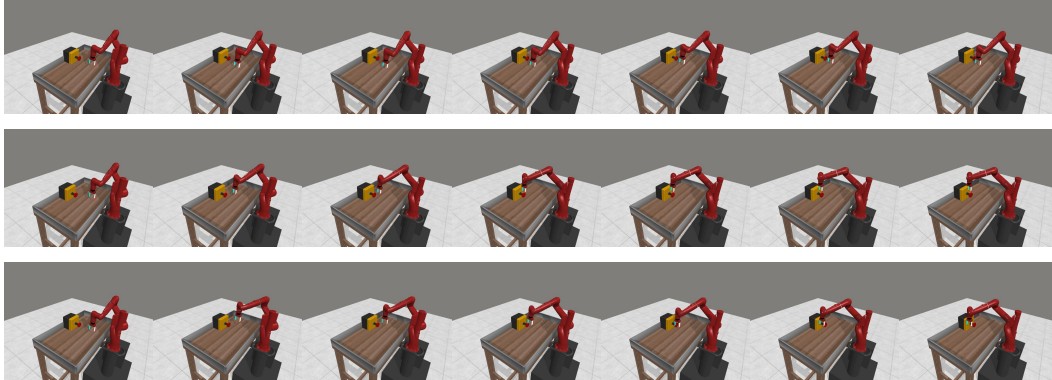

(b) LLM-generated trajectories that are not task-successful but still semantically meaningful and task-aligned. Although these examples fail to press the button, they exhibit structured behaviors such as approaching the button and making plausible press attempts.

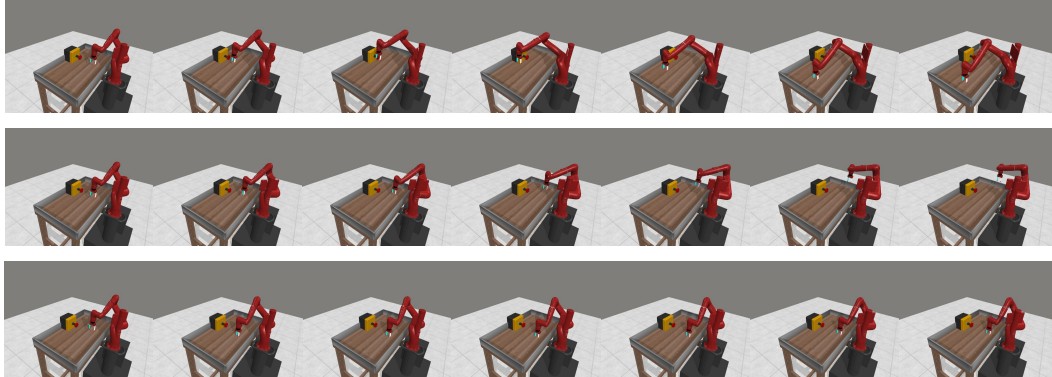

(c) Early-stage random exploration trajectories, which are typically noisy and unstructured. Compared to these, even non-successful LLM-generated trajectories can serve as preference anchors due to their goal-oriented structure.

Figure 19: Examples of LLM-generated bootstrap trajectories from foresight generation, compared with random trajectories from early-stage policy rollout.

We observe that the LLM-generated trajectories exhibit two typical patterns. In some cases, as shown in Fig. 19a, they successfully complete the task while showing meaningful diversity in execution, such as varying the robot's starting position or adjusting the gripper height when pressing the button. In other cases, as shown in Fig. 19b, the generated trajectories fail to complete the task but still demonstrate semantically aligned and structured behaviors, including approaching the button and attempting to press it in a plausible manner.

In contrast, as shown in Fig. 19c, early-stage random exploration trajectories are generally unstructured and uniformly poor, lacking coherence or goal-directed motion. Notably, even the unsuccessful but structured LLM-generated samples serve as useful preference anchors when paired with these random trajectories. When used in combination with uncertainty-based query selection, such pairings are more likely to be selected for labeling due to their high predictive disagreement across the preference ensemble. This leads to more informative preference queries in the early stages of reward model training.

## G.4    Qualitative Analysis of the Hindsight Trajectory Augmentation Module

To further illustrate how the hindsight trajectory augmentation module operates, we visualize two counterfactual variants of a preferred trajectory, as shown in Fig. 20. These examples demonstrate how the LLM generates counterfactual samples by performing minimal interventions at key causal steps.

In Fig. 20a, the LLM introduces a delay by holding the object longer before releasing it, simulating a temporally adjusted strategy while preserving the rest of the trajectory. In Fig. 20b, the LLM adds a brief hesitation before grasping the object, representing a minimal behavioral perturbation that weakens the preference without altering the trajectory's overall structure. By applying such targeted, minimal edits to causally critical steps, the preference difference becomes more sharply attributed to specific actions or states, effectively isolating the cause of preference reversal. This sharpens the causal signal and provides more informative supervision for the reward model, thereby improving temporal credit assignment and helping the model learn which precise behaviors lead to better or worse preferences.

**Preferred Trajectory**

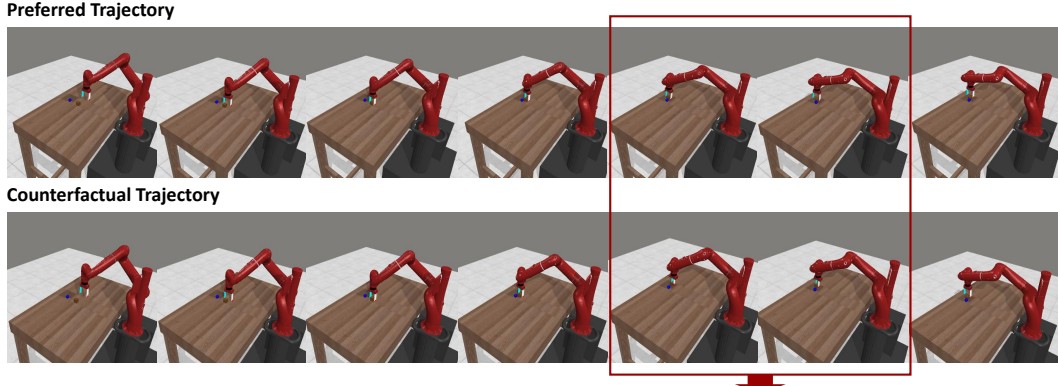

**Counterfactual Trajectory**

*Hold the cube for a longer period of time before releasing it to the target area*

(a) LLM generates a counterfactual trajectory by adding a delay: holding the cube for a longer period of time before releasing it to the target area, while other steps remain the same as in the preferred trajectory.

**Preferred Trajectory**

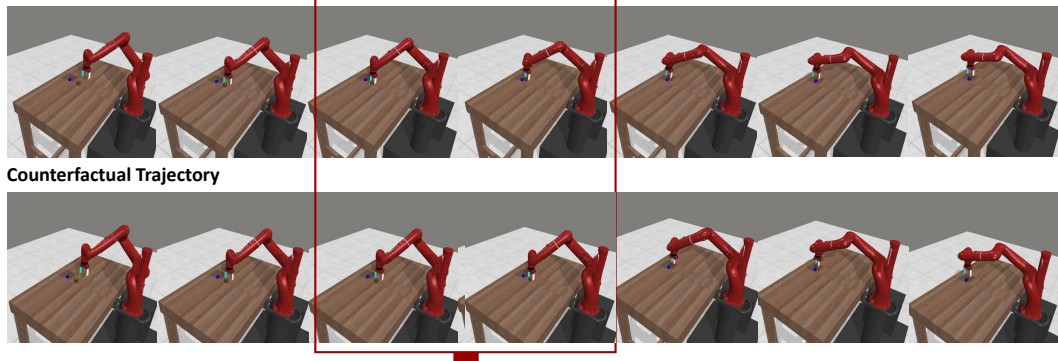

**Counterfactual Trajectory**

*Add a short hesitation before grasping the cube*

(b) LLM generates a counterfactual trajectory by inserting a brief hesitation before grasping the cube, while keeping the other steps identical to those in the preferred trajectory.

Figure 20: Examples of LLM-generated counterfactual trajectories based on minimal intervention at causally critical steps.

## G.5 Policy Visualizations of Different Methods

To gain more insights into how different methods shape policy behaviors, we visualize the robot behaviors learned by PRIMT and the best-performing baseline on several representative tasks. The visualizations are shown in Fig. 21 and Fig. 22. We observe that PRIMT leads to more efficient and coherent manipulation and locomotion behaviors, demonstrating better task completion strategies and smoother trajectories compared to the baseline.

**RL-SaLLM-F**

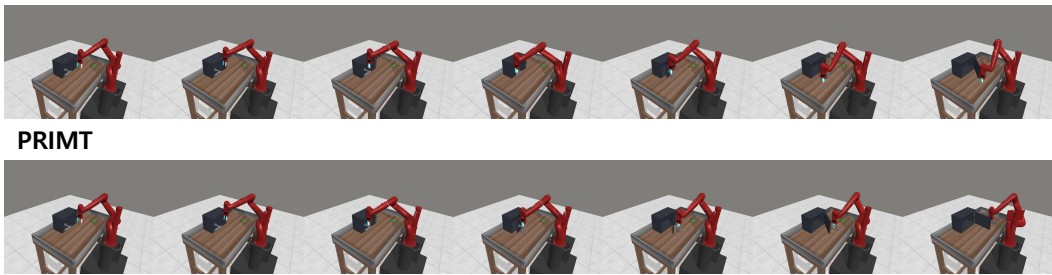

**PRIMT**

(a) Policy Visualizations between PRIMT and RL-SaLLM-F on the Door Open task.

**PrefCLM**

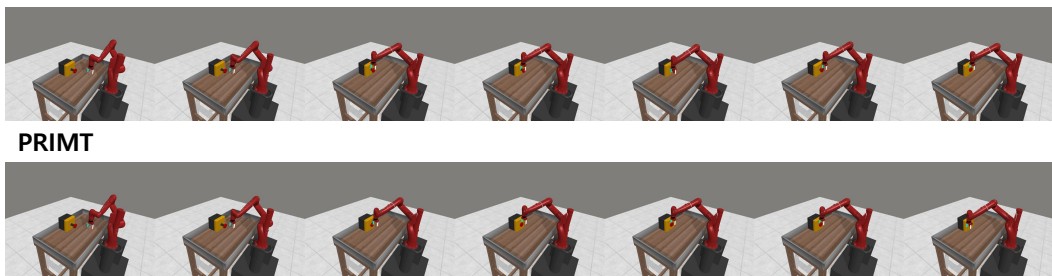

**PRIMT**

(b) Policy Visualizations between PRIMT and PrefCLM on the Button Press task.

**PrefCLM**

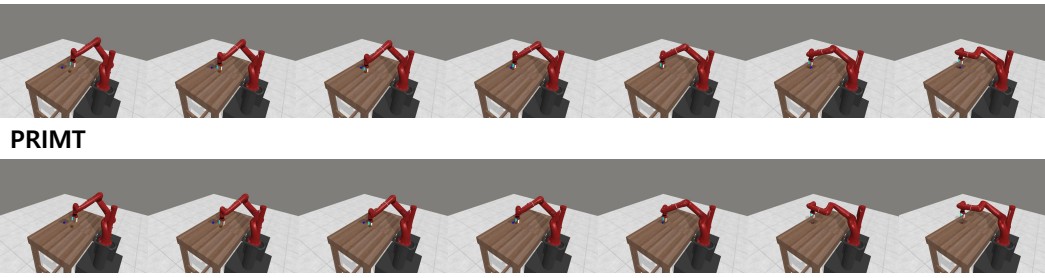

**PRIMT**

(c) Policy Visualizations between PRIMT and PrefCLM on the Sweep Into task.

Figure 21: Examples of policy visualization between different methods in MetaWorld.

## G.6 Preliminary Experiments on a Bimanual Manipulation Task

We further conducted preliminary experiments on a more difficult bimanual manipulation task, TwoArmPegIn-Hole from RoboSuite. As shown in Table 4, PRIMT still significantly outperforms the baseline RL-VLM-F and approaches the PbRL oracle PrefGT. These results suggest that PRIMT potentially generalizes well to more complex, high-dimensional settings.

Table 4: Preliminary results on the *TwoArmPegInHole* task: success rates (%) during training.

| Method | Max SR (%) | Mean SR (%) | Final SR (%) |
|---|---|---|---|
| PrefGT | 78.40 | 67.17 | 78.03 |
| RL-VLM-F | 32.66 | 25.48 | 32.66 |
| PRIMT | 65.06 | 53.26 | 64.15 |

**RL-SaLLM-F**

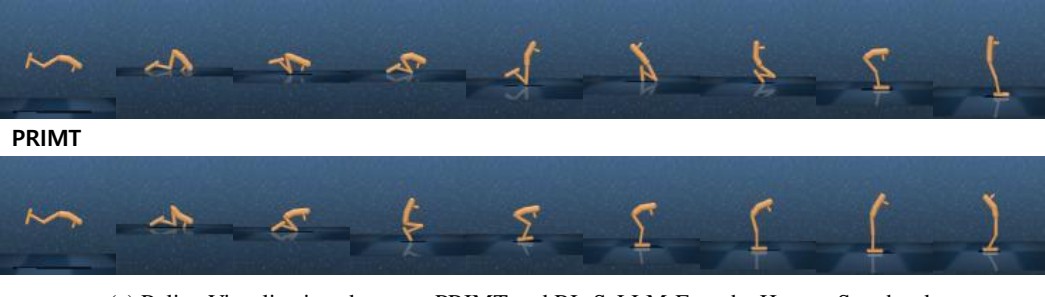

**PRIMT**

(a) Policy Visualizations between PRIMT and RL-SaLLM-F on the Hopper Stand task.

**RL-SaLLM-F**

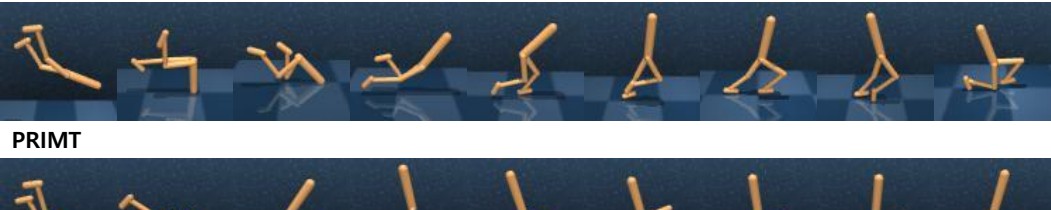

**PRIMT**

(b) Policy Visualizations between PRIMT and RL-SaLLM-F on the Walker Walk task.

Figure 22: Examples of policy visualization between different methods in DeepMind Control.

## G.7 Comparison with Dense-Reward RL

To further evaluate the effectiveness of our method, we introduce a new baseline GT, an RL policy trained using ground-truth dense rewards provided by the benchmark environments and SAC. This baseline enables a direct comparison between PRIMT, preference-based learning, and conventional dense-reward RL. Learning curves are reported in Figure 23, which summarizes the final success rates and episode returns of GT, PrefGT, the best-performing baseline, and PRIMT across all tasks.

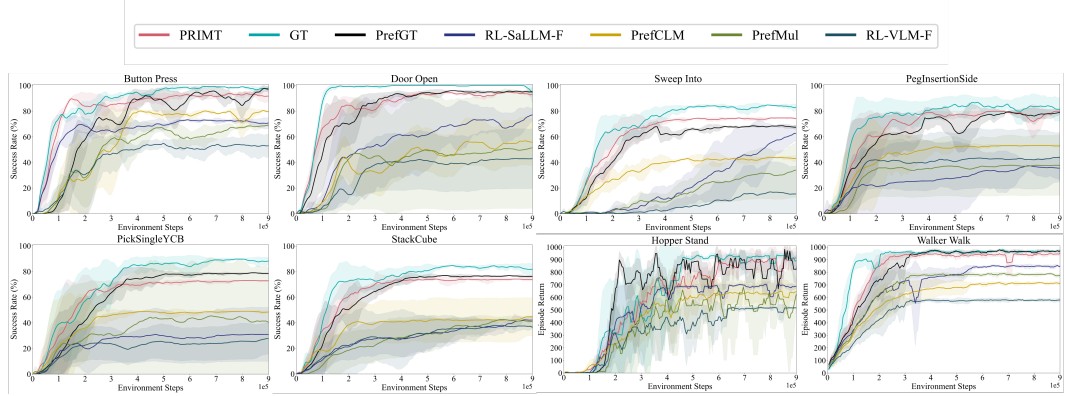

Figure 23: Learning curves including the ground-truth dense-reward RL baseline.

The results indicate that GT (dense-reward RL) achieves the highest performance across all tasks, as expected due to its access to dense, step-level supervision. Nonetheless, PRIMT attains between 1.9% and 17.3% of the performance gap relative to GT, demonstrating that PRIMT effectively approximates the behavior of dense-reward RL without requiring explicit reward shaping. This highlights PRIMT's scalability and efficiency in scenarios where reward engineering or extensive human feedback is infeasible.

### G.8 Qualitative Reward Alignment Analysis

To qualitatively examine the relationship between learned rewards and the ground-truth task rewards shown in Figure 4, we performed a quantitative analysis using the $R^2$ coefficient to measure reward alignment across different methods. The baseline RL-VLM-F was selected as the reference for comparison. The computed $R^2$ values for each task and method are summarized in Table 5.

Table 5: $R^2$ Coefficient Analysis (Reward Alignment with Ground Truth).

| Task | PRIMT | w/o CauAux | w/o HindAug | RL-VLM-F |
|------|-------|-----------|-------------|----------|
| PegInsertionSide | 0.56 | 0.28 | 0.23 | 0.37 |
| PickSingleYCB | 0.84 | 0.01 | 0.34 | -0.05 |
| StackCube | 0.78 | -1.31 | -2.28 | -1.50 |
| ButtonPress | 0.87 | 0.68 | 0.53 | -0.61 |
| DoorOpen | 0.64 | -1.19 | 0.15 | -4.72 |
| SweepInto | 0.88 | 0.83 | 0.73 | -0.27 |
| WalkerWalk | 0.33 | 0.19 | 0.02 | -2.29 |

The results show that PRIMT achieves consistently higher $R^2$ coefficients compared to other variants and the baseline RL-VLM-F, indicating stronger alignment between its learned reward signals and ground-truth task rewards. Both the hindsight augmentation and causal auxiliary loss components contribute positively to this improvement, suggesting their importance in enhancing credit assignment quality during reward learning.

### G.9 Real-world Experiments

We further evaluate our method in the real world on two tasks from the Robosuite benchmark [64]: *Block Lifting* and *Block Stacking*. In the Block Lifting task, the goal is to control a robotic arm to lift a cube to a certain height. In the Block Stacking task, the objective is to place one cube on top of another. We adopt PrefCLM [15] as a real-world baseline.

Policies are first trained in simulation using Robosuite [64], and then deployed on a physical Kinova Jaco2 robotic arm. We leverage RoboSuite's comprehensive sim2real capabilities, which include dynamics randomization and sensor modeling with realistic sampling rates, delays, and noise corruption.

During training, we closely matched the robot model, camera configuration, and workspace setup with the real hardware. The control frequency and action space were also kept consistent. To ensure safety during deployment, we imposed a soft constraint: if the turning angle or acceleration exceeded a threshold, the corresponding action was discarded. Performance results are shown in Figure 24. We evaluated each model over 10 randomized trials (e.g., varying initial block positions). PRIMT achieved 7/10 successful lifts and 6/10 successful stacks, whereas the PrefCLM baseline achieved 4/10 and 2/10 successes, respectively. More videos of real-world experiments are on our project website.

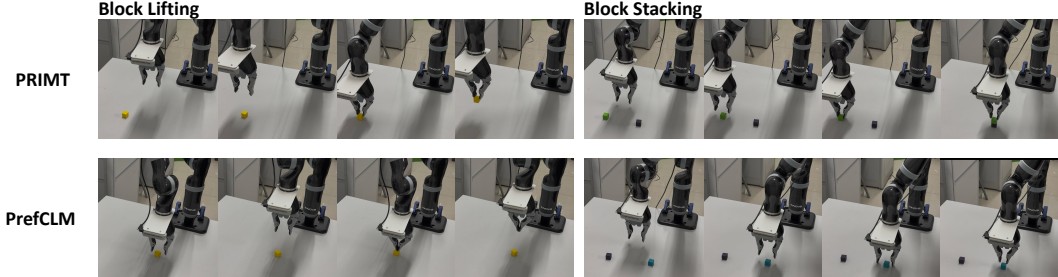

Figure 24: Demos of real-world manipulation experiments on the Block Lifting and Block Stacking tasks.

## H  Discussion on Assumptions, Limitations, and Broader Impact

### H.1  Assumptions and Limitations

In this work, we make the following assumptions, each grounded in prior work and empirical observations: 1) We assume that foundation models (FMs), including LLMs and VLMs, are pretrained on large-scale and diverse

corpora, enabling generalization across a wide range of environments and tasks; 2) We assume that VLMs are capable of analyzing sequences of visual inputs to extract spatial and semantic cues, while LLMs can interpret temporal structures, reason over structured inputs, and generate coherent task-related feedback; 3) We assume that LLMs, when guided by structured prompts and domain-aligned vocabulary, are capable of generating or editing robot trajectories in a coherent and task-relevant manner. This assumption builds upon recent work demonstrating the ability of LLMs to plan actions [41], generate code-based controllers [38], and perform trajectory synthesis via natural language [36]; and 4) We assume that task progress can be inferred at least partially through either visual sequences (e.g., keyframe differences, spatial transitions) or textual descriptions (e.g., arrays of actions or states).

Our work is fundamentally built upon FMs, and therefore, like other FM-based methods, it inherits certain dependencies on the capabilities and limitations of these models. While this reliance is inherent to the design, we have taken steps to mitigate it: for example, leveraging multimodal feedback to enhance the quality of preference signals and adopting a structured code-generation paradigm to improve the quality of generated trajectories. As shown in our ablation study on LLM backbone selection, PRIMT maintains reasonable performance even when using a weaker model (`gpt-4o-mini`), indicating robustness to FM capacity.

Another limitation lies in the cost associated with FM usage. Specifically, our multimodal feedback generation, fusion module, and bidirectional trajectory synthesis involve multiple FM calls. To provide a clearer picture of the resource requirements, we present a detailed resource usage comparison in Table 6 and the corresponding cost-performance trade-offs in Table 7.

We observe that compared to the RL-VLM-F and Se-LLM-F baselines, cost and training time increased moderately (by 38–47% and 30–69%, respectively). While this increase is notable, performance gains were substantial (+19–117%), resulting in efficiency improvements of $2.0\times$ and $1.4\times$, respectively. This justifies the additional FM usage in PRIMT, given the performance benefits are notable. More importantly, compared to the performance achieved by collecting human feedback (represented by PrfSlt) where expert-sourced teachers provide ground truth preferences, PRIMT achieves comparable performance (within 1–3%) while reducing estimated human annotation costs by over 92%. (based on 0.05–0.1 per preference label for 20,000 queries on platforms such as Prolific and MTurk). This highlights PRIMT's scalability and practicality as a paramount step to expensive human-in-the-loop methods. Therefore, we believe PRIMT strikes a good balance between performance and cost-effectiveness, providing a practical path toward scalable preference learning.

Table 6: Resource usage comparison across different methods and environments

| Method | Average Usage Cost ($) _MetaWorld / ManiSkill / DMC_ | Average Training Time (h) _MetaWorld / ManiSkill / DMC_ |
|---|---|---|
| **RL-VLM-F** | 84.14 / 57.42 / 84.27 | 4.3 / 5.1 / 4.5 |
| **Sa-LLM-F** | 83.22 / 55.31 / 89.69 | 5.2 / 5.6 / 5.2 |
| **PRIMT** | 120.42 / 79.73 / 124.01 | 6.8 / 7.3 / 7.6 |
| **Human** | 1,000–2,000 | N/A |

*- FM cost estimated based on GPT-4o API pricing at the time of experiments*
*- Human cost estimated from 20,000 preference queries for a task at $0.05–$0.10 per label on platforms like Prolific and MTurk*

Table 7: Cost-performance trade-off comparison of PRIMT against baseline methods

| Baseline | Cost | Time | Performance | Efficiency[†] |
|---|---|---|---|---|
| | _MetaWorld / ManiSkill / DMC_ | | | |
| **vs RL-VLM-F** | +43%/+39%/+47% | +58%/+43%/+69% | +95%/+117%/+68% | $2.0\times$ |
| **vs Sa-LLM-F** | +45%/+44%/+38% | +31%/+30%/+46% | +32%/+109%/+19% | $1.4\times$ |
| **vs Human (PrefGT)** | −92%/−95%/−92% | —/—/— | −1%/−%/−% | $47\times$ |

*- [†] Efficiency = Average performance gain / Average resource increase*
*- Performance is measured using the final return from the learning curves presented in Figure 2*
*- Values shown are relative changes across MetaWorld / ManiSkill / DMC environments, respectively*

## H.2 Impact Statement

This work explores the integration of foundation models into preference-based reinforcement learning (PbRL), which aims to improve learning efficiency and robustness through multimodal feedback and trajectory synthesis. By leveraging VLMs and LLMs, our approach reduces the need for extensive human supervision, potentially broadening access to PbRL techniques in domains with limited annotation resources.

However, the use of FMs raises concerns around data privacy, transparency, and decision-making fairness. As with other FM-based methods, our system may inherit unintended biases from pretrained models, which could impact downstream behavior. We encourage future work to explore bias mitigation, model auditing, and safe deployment strategies, especially in high-stakes or safety-critical applications.

Overall, our work offers a promising step toward scalable and generalizable robot learning systems. Moreover, the underlying principles, multimodal feedback fusion, and trajectory-level reasoning, can extend beyond robotics to broader sequential decision-making scenarios in fields such as education, healthcare, and interactive agents.

