# OpenReview forum: "PRIMT: Preference-based Reinforcement Learning with Multimodal Feedback and Trajectory Synthesis from Foundation Models"
_NeurIPS.cc/2025/Conference — NeurIPS 2025 oral_

### Official Review · Reviewer_397q · 2025-06-23

**Clarity:** 2
**Significance:** 3
**Originality:** 2
**Rating:** 4
**Confidence:** 4

**Summary:**

This submission introduces PRIMT, a preference-based RL framework. It uses foundation models (LLMs and VLMs) to annotate preference labels of trajectories and synthesize both foresight and hindsight counterfactual trajectories. All these efforts are for better learning reward models. RL trained with the learned reward models demonstrates faster convergence and better asymptotic performance over several baseline methods on three benchmarks, ranging from locomotion to robot arm manipulation. Real-world deployment is also provided.

**Questions:**

- What is the exact implementation of the reward model? How did you train that?
- What is the RL algorithm used?
- Fig 4b, what is the blue curve for?
- Fig 4b, it would be more informative to quantify reward alignment. Could consider metrics such as R2 coefficient.
- Sec. 4.5, real-world deployment, do you perform sim2real transfer? Is it state based or image based? Or do you simply replay trajectories collected in sim?

**Ethical Concerns:**

["NO or VERY MINOR ethics concerns only"]

**Final Justification:**

Most of my concerns have been addressed in authors' response. I'd like to keep my original positive rating.

**Limitations:**

Limitations not discussed.

**Paper Formatting Concerns:**

No major paper formatting concerns.

**Quality:**

3

**Strengths And Weaknesses:**

## Strengths
- The submission is well written and clearly presented.
- The intuition of using multiple modalities for preference-based RL makes sense.
- Experiment evaluation and ablation studies are comprehensive.
- The proposed method performs quite good.

## Weaknesses
- While the proposed method is impressive, it is a system consisting of many existing techniques. It's unclear how the proposed method compared to others at a system level.
- While authors claim PbRL alleviates reward engineering burden, demonstrated tasks are still largely simple. It would be beneficial to see if the proposed method can work on more complicated robotic tasks, such as mobile manipulation, or long-horizon tasks such as MineRL where reward is sparse.
- Authors compare against several PbRL methods. It would be more informative to see how the proposed method compare against RL trained directly on ground-truth dense reward (if exist), or imitation learning agents trained with demonstrations.
- Runtime complexity and required compute not discussed. I imagine the proposed method would take longer time and consume more compute than baseline methods.
- Missing some details, see questions below.
- No limitation discussed.

---

> ### Author Rebuttal · Authors · 2025-07-31
>
> # Response to Reviewer 397q
>
> **Dear Reviewer 397q,**
>
> Thank you for your encouraging feedback on the clarity of our presentation, the motivation behind our methodology, and the rigor of our experiments. We are glad to have the opportuniy to repsond to your insightful comments as below.
>
> ---
> ### **W1: System Composition**
>
> We appreciate this thoughtful comment and would like to clarify that while PRIMT builds upon emerging directions in FM-based learning (e.g., using foundation models for preference labeling and trajectory synthesis), each component is **specifically designed to address distinct, well-motivated challenges** in PbRL and introduces **novel architectural elements**.
>
> For example, to our best knowledge, we are **the first to propose a multimodal evaluation setting** in PbRL, and introduce a **neuro-symbolic fusion mechanism** to overcome feedback inconsistency in existing single-modality approaches. We also propose **bidirectional trajectory synthesis** to tackle early-stage query ambiguity and poor credit assignment—introducing, for the first time in PbRL, **counterfactual reasoning coupled with a causal auxiliary loss** for fine-grained credit attribution.
>
> Regarding **system-level comparisons**, we conduct explicit evaluations between PRIMT and several **full-system baselines**, all of which support complete PbRL pipelines and represent distinct system configurations:
>
> - RL-VLM-F: a unimodal system using only VLMs for feedback.
> - RL-SaLLM-F: a unimodal LLM-based system augmented with foresight trajectory generation.
> - PrefCLM: a crowd-sourced unimodal fusion method.
> - PrefMul: our constructed naïve multimodal system.
> - PrefGT: a human-in-the-loop PbRL upper bound using scripted expert preferences.
>
> As shown in Section 4 of the manuscript, PRIMT consistently outperforms FM-based baselines, resulting in **non-trivial gains at the system level**.
>
> Additionally, following your suggestion, we include a **new RL baseline with dense rewards** for additional reference. Please see our response to your **W3** for details. This comparison further highlights that PRIMT can be even competitive with dense-reward performance in some tasks, despite relying only on **FM-based zero-shot preference supervision**.
>
> ---
>
>
> ### **W2: Task Complexity**
>
> Our task selection **follows established practices in the PbRL literature**, where MetaWorld and DMC are widely adopted as standard benchmarks for preference-based methods. To further expand the evaluation scope, we have already incorporated ManiSkill, **a newer and more challenging benchmark** featuring complex object interactions and high-dimensional control.
>
> PbRL itself is still an emerging field, and even human-in-the-loop PbRL algorithms currently face limitations when applied to highly complex or long-horizon tasks. That said, we conducted **preliminary experiments on a more difficult bimanual manipulation task**, *TwoArmPegInHole* from RoboSuite. As shown in Table 3, PRIMT still significantly outperforms the baseline RL-VLM-F and approaches the PbRL oracle PrefGT. These results suggest that PRIMT potentially generalizes well to more complex, high-dimensional settings.
>
> In the revision, we will include this result and expand the discussion section to provide further insights on extending PRIMT to more complex and long-horizon robotic tasks.
>
> ---
>
> **Table 3: Preliminary results on the TwoArmPegInHole Task in Success Rate during training**
> | Method      | Max SR | Mean SR | Final SR |
> |-------------|--------|---------|-----------|
> | PrefGT      | 78.40  | 67.17   | 78.03     |
> | RL-VLM-F    | 32.66  | 25.48   | 32.66     |
> | PRIMT       | 65.06  | 53.26   | 64.15     |
>
>
> ---
>
> ### **W3: Comparison with Dense-Reward RL**
>
> Following your suggestion, we added a new baseline **GT** (RL policy trained using ground-truth dense rewards provided by the benchmark environments and SAC) to provide additional comparison insights.
>
> Unfortunately, due to the new rebuttal policy, we cannot provide updated learning curves. As an alternative, we extract key performance data in Table 4, showing the final success rates/episode returns of GT, PrefGT, the best performing baseline, and PRIMT across all tasks.
>
> ---
>
> **Table 4: Final Performance (Success Rate % / Episode Return) Across Tasks**
>
> | Task               | GT (Dense RL) | PrefGT (Oracle PbRL) | PRIMT (Ours) | Best Baseline |
> |--------------------|---------------|----------------------|--------------|---------------|
> | PegInsertionSide   | 82.38         | 77.65                |79.63    | 53.28         |
> | PickSingleYCB      | 88.14         | 78.89                | 72.85        | 49.00         |
> | StackCube          | 82.51         | 76.64                | 73.95        | 44.73         |
> | ButtonPress        | 97.60         | 95.67                | 93.21        | 80.19         |
> | SweepInto          | 84.15         | 68.38                | 74.15   | 66.87         |
> | DoorOpen           | 100.00        | 98.10               | 95.00        | 74.00         |
> | WalkerWalk         | 970.89        | 969.44               | 937.58       | 854.59        |
> | HopperStand        | 924.09        | 909.65               | 906.58       | 704.89        |
>
> ---
>
> As expected, the results reveal that GT (dense-reward RL) consistently outperforms all PbRL methods, even PrefGT (oracle PbRL), due to its access to dense, step-level supervision. However, we observe that **PRIMT reaches within 1.9%–17.3% of the performance of RL policies trained with ground-truth dense rewards** across a variety of tasks. This result further highlights the potential of PRIMT towards scalable and efficient PbRL without requiring hand-crafted dense rewards or human feedback, reinforcing its practicality in real-world settings where reward engineering is difficult or infeasible.
>
> ---
> ### **W4: Computational Cost**
>
> Thank you for highlighting this important practical consideration. Due to its multimodal architecture, PRIMT does involve higher computational overhead compared to other FM-based baselines. To provide a more transparent analysis, we have compiled a detailed comparison of compute costs and efficiency trade-offs, presented in **Tables 1 & 2** in our response to Reviewer r5ng.
>
> Compared to existing FM-based methods (RL-VLM-F and Sa-LLM-F), PRIMT increases **FM Usage costs by 38–47%** and **training duration by 30–69%**. However, this moderate cost increase yields **substantial performance gains of 19–117%**, resulting in **overall efficiency improvements of 1.4–2.0×**.
>
> Lastly, as discussed in Appendix G.2, PRIMT can also **run with lighter foundation models** (e.g., GPT-4o-mini), achieving 94% cost savings at a 16–26% drop in performance—still yielding strong cost-effectiveness and adaptability for lower-resource settings.
>
> We will incorporate this analysis into the final version to more clearly situate PRIMT’s computational demands within the broader cost-performance landscape of PbRL methods.
>
> ---
>
> ### **W5 & Q1-Q2: Implementation Details**
>
> First, we want to apologize that due to page limitations in the current version, we could not include all necessary implementation details in the main paper. The comprehensive experimental details are provided in **Appendix F** of current manuscript.
>
> As noted in Section F.2&3, we use **PEBBLE as the PbRL backbone**. Following the standard PEBBLE design, we adopt a 3-layer ensemble architecture for the reward model and SAC as the RL algorithm.
>
> ---
> ### **W6: Discussion of Limitation**
>
> We would like to clarify that a detailed discussion of limitations is already included in **Appendix H.1** of the current submission. We apologize if this was not sufficiently visible, and will move a condensed version of the limitations discussion into the main paper in the revision for improved clarity.
>
> ---
>
> ### **Q3 & Q4: Figure 4b**
>
> Thank you for pointing out the confusion. The **blue curve in Figure 4b represents the baseline RL-VLM-F**. We will update the legend of Figure 4b for clarity.
>
> Thanks for the excellent suggestion of using **R² coefficient to quantify reward alignment**. Below we present the R² coefficients comparing learned rewards to ground-truth task rewards:
>
> ---
>
> **Table 5: R² Coefficient Analysis (Reward Alignment with Ground Truth)**
>
> | Task | PRIMT | w/o.CauAux | w/o.HindAug | RL-VLM-F |
> |------|-------|------------|-------------|----------|
> | PegInsertionSide | 0.56 | 0.28 | 0.23 | 0.37 |
> | PickSingleYCB | 0.84| 0.01 | 0.34 | -0.05 |
> | StackCube | 0.78 | -1.31 | -2.28 | -1.50 |
> | ButtonPress | 0.87 | 0.68 | 0.53 | -0.61 |
> | DoorOpen | 0.64| -1.19 | 0.15 | -4.72 |
> | SweepInto |0.88 | 0.83 | 0.73 | -0.27 |
> | WalkerWalk |0.33 | 0.19 | 0.02 | -2.29 |
>
> ---
>
> These results further highlight that both hindsight augmentation and our causal auxiliary loss significantly **improve the quality of credit assignment during reward learning**. We will integrate this analysis into the paper to provide a more quantitative perspective on alignment quality.
>
> ---
> ### **Q5: Real-World Deployment**
>
> Thank you for the question. Policies were trained in RoboSuite simulation using state-based observations and then deployed on the physical Kinova Jaco robot. We leverage RoboSuite's comprehensive sim2real capabilities, which include dynamics randomization and sensor modeling with realistic sampling rates, delays, and noise corruption.
>
> During training, we closely matched the robot model, camera configuration, and workspace setup with the real hardware. The control frequency and action space were also kept consistent. To ensure safety during deployment, we imposed a soft constraint: if the turning angle or acceleration exceeded a threshold, the corresponding action was discarded.
>
> We will clarify the real-world deployment protocol in more detail in Appendix G.6 of the revised version.
>
> ---
>
> Thank you again for taking the time to review our paper and rebuttal. We hope our responses could address your concerns.

---

### Official Review · Reviewer_87iK · 2025-06-30

**Clarity:** 3
**Significance:** 3
**Originality:** 3
**Rating:** 5
**Confidence:** 4

**Summary:**

This paper introduces a framework for Preference-based Reinforcement Learning (PbRL), called PRIMT, from Foundation Model (FM) feedback. It addresses two core challenges: (i) the ambiguity of early-stage feedback due to low-quality trajectories at the beginning of training and (ii) the difficulty of credit assignment in the case of trajectory feedback. To improve feedback quality, PRIMT combines trajectory-level evaluations from LLMs and VLMs using a probabilistic soft logic approach. This "fusion" process incorporates confidence scores, cross-modal agreement, and trajectory context to resolve conflicts. To handle sparse and ambiguous training data, PRIMT also uses LLMs to generate synthetic trajectories in the form of "foresight" generation (populating a replay buffer) and "hindsight" generation (edits to create counterfactuals). The authors evaluate and compare PRIMT on various simulated robotics tasks and two real-world robot demos. They also conduct ablations to quantify the impact of each component.

**Questions:**

1. How often do "hindsight" edits yield invalid or infeasible actions, and how do you filter them? The paper mentions that LLMs produce "plausible" edits, but there’s no quantitative analysis of failure rates or their impact on learning stability.
2. As far as I can tell, you prompt the FMs to return a "confidence" score via chat templates. How realiable is that really? Are there other alternatives, e.g., using log-probabilities from the LLM or maybe other uncertainty quantification methods?

**Ethical Concerns:**

["NO or VERY MINOR ethics concerns only"]

**Final Justification:**

In my opinion, this is a good paper. One of the biggest strengths and a key reason for my positive assessment is the extensive experimental evaluation. The experiments include an unusually large amount of different tasks and various ablations.

**Limitations:**

Yes.

**Paper Formatting Concerns:**

None.

**Quality:**

4

**Strengths And Weaknesses:**

#### Strengths:

1. The experimental evaluation in the paper is extensive and thoroughly performed. Various ablations are provided which anticipated many of the questions I had when reading the paper. The chosen baselines and the experimental setup make sense and strongly support the paper's claims. The evlauation also covers an exceptional breadth of test scenarios including DMC, MetaWorld, Maniskill and a real-world example. Additional metrics such as label accuracy w.r.t. a proxy oracle feedback model are also provided and ablated.
2. The structured fusion of VLM and LLM feedback using probabilistic soft logic is interesting and shows significant empirical improvements over using either modality alone. Similarly, the problems of query ambiguity and credit assignment are clearly motivated and convincingly addressed using the proposed "foresight" trajectory generation and the "hindsight" trajectory edits.
3. The presentation of the methodology, the related work, and the explanations of the proposed approach are detailed and clear.
4. I also appreciated the extensive section on the paper's assumptions and limitations in Appendix H.

#### Weaknesses:

1. My biggest concern is that much of the methodology feels like "stacking heuristics" and informed guesswork. This is probably expected (to some extent) when working with FM-based feedback and aiming for better empirical performance in complex robotics tasks. My concern is slightly alleviated by the extensive experimental evaluation and many ablations, which also lead to my overall positive assessment of the paper despite the above.
2. The cost associated to the extensive use of FMs is a natural drawback of the proposed approach. Here, it would have been great to see the total number of FM queries used per experiment, the cost associated with them, and similar metrics. If these are available, I recommend to report them in the paper.

---

> ### Author Rebuttal · Authors · 2025-07-30
>
> # Response to Reviewer 87iK
>
> **Dear Reviewer 87iK,**
>
> Thank you for your thorough evaluation and positive assessment of our methodology and experiments. We sincerely appreciate your recognition of our contributions and are glad to address your insightful questions below.
>
> ---
>
> ### **W1: "Stacking Heuristics" Concern**
>
> Thank you for raising this thoughtful point. We understand the concern regarding modular complexity due to the multi-component nature of our method, and we appreciate your acknowledgment that our extensive ablations help alleviate this issue.
>
> Here, we would like to further clarify our design rationale. PRIMT is not simply a set of stacked heuristics, but rather a **principled, problem-driven framework** where each module is designed to address a specific, empirically observed challenge in FM-driven preference learning.
>
> Our design process began with the observation (Appendix A) that **LLMs and VLMs exhibit distinct limitations yet complementary strengths** (Figure 8), motivating multimodal feedback fusion. We further identified **context-dependent performance patterns** across modalities, leading to the adoption of Probabilistic Soft Logic (PSL) for more interpretable and robust integration. Finally, we found that even high-quality synthetic feedback could not resolve **inherent challenges in PbRL**: query ambiguity and credit assignment, which we address through foresight and hindsight generation.
>
> These components are not standalone heuristics but are mutually complementary, forming an integrated system where fusion improves label quality, foresight provides early-stage preference anchors, and hindsight enhances fine-grained credit assignment—together aiming to facilitate a generalizable and robust zero-shot PbRL framework.
>
> We will revise the main paper to make this design motivation clearer and more explicit.
>
> ---
>
> ### **W2: Extensive FM Usage**
>
> We appreciate the opportunity to provide more insights on the resource usage of PRIMT. Due to its multi-agent architecture, PRIMT incurs higher FM usage compared to prior FM-based baselines. To provide a more transparent analysis, we have compiled a detailed comparison of **compute costs and efficiency trade-offs**, presented in **Tables 1 & 2** in our response to Reviewer r5ng.
>
> Compared to FM-based baselines (RL-VLM-F and Sa-LLM-F), PRIMT increases FM usage cost and training time **moderately** (by 38–47% and 30–69%, respectively), but yields **substantial performance improvements** (+19–117%), resulting in **efficiency gains of up to 2.0×**. We believe this could justify the added cost.
>
> More importantly, when compared to collecting **real human preference labels**—represented by PrefGT where expert scripted teachers provide ground-truth preferences—PRIMT achieves **comparable performance (within 1–3%) while reducing estimated annotation costs by over 92%** (based on $0.05–$0.10 per label for 20,000 queries on platforms like Prolific or MTurk). This suggests PRIMT offers a scalable and cost-effective alternative to traditional human-in-the-loop methods.
>
> Furthermore, our ablation in Section G.2 demonstrates that PRIMT remains effective even with lower-cost FMs: using GPT-4o-mini yields 94% cost savings with a 16–26% drop in performance, resulting in a **13× improvement in cost-performance ratio**. This highlights the framework’s **flexibility for resource-constrained scenarios**.
>
> We will integrate this analysis into the revised version to better contextualize PRIMT’s scalability and cost-efficiency.
>
> ---
>
> ### **Q1: Quality of Hindsight Trajectory Samples**
>
> To ensure the quality and informativeness of counterfactual trajectories generated during hindsight augmentation, we apply a two-step filtering process:
>
> - We compute the L1 distance between the edited state-action pairs and their original counterparts, retaining only those with minimal deviation to ensure the counterfactual reflects a plausible, fine-grained edit.
>
> - The filtered counterfactuals are then passed through the LLM-based intra-modal fusion module to verify that they are indeed less preferred than the original.
>
> These ensure the final counterfactuals support the loss in Eq. 15 for meaningful credit assignment during reward model training.
>
> However, we did observe that not all counterfactuals strictly adhere to physical constraints—for example, some edits introduce abnormal accelerations or sharp turns to make a trajectory appear less preferred.
>
> We initially considered adding a separate physical feasibility filter, such as VLM-based rendering evaluation or simulation-based rollouts (as in [1]). However, we ultimately decided against it for several principled reasons:
>
> - First, as established in recent work [2], **generated trajectories do not need to adhere to strict physical constraints**, as they serve solely for preference comparisons to train the reward model, rather than for extracting policies, rollong out in real-world or constructing a world model.  Given that our **reward model is Markovian**—with immediate rewards depending only on the current state—physically implausible transitions do not undermine reward learning, as long as the local preference signal remains valid.
>
> - We also conducted **pilot experiments** and found that our hindsight trajectory generation aligns well with this principle—despite occasional infeasible transitions, the generated edits consistently support effective reward learning.
>
>
> That said, we also observed that our chain-of-thought trategy for generating counterfactuals encourages more coherent and feasible edits compared to directly prompting LLMs to generate full alternative trajectories (as used in the baseline RL-SaLLM-F).
>
> We will clarify these design choices and provide additional insights and examples in the revised version.
>
> ---
>
> ### **Q2: Confidence Estimation**
>
> Estimating preference confidence plays a critical role in our multimodal feedback fusion pipeline. We explored several popular methods [3], including logit-based/entropy-based uncertainty, chain-of-thought (CoT) self-evaluation, repeated sampling, and external critic models.
>
> However, our design goal was to maintain general applicability across both open- and closed-source foundation models, as well as to reduce inference overhead. Therefore, we avoided logit-based or entropy-based metrics, which are often unavailable in API-based FM deployments. We also excluded external evaluators, which would significantly increase cost.
>
> Finally, we adopted a dual-component confidence estimation (as shown in Eqs. 4-5) that combines complementary signals:
>
> - Internal certainty: Average self-reported confidence from predictions that align with the final consensus decision
> - Behavioral consistency: Agreement ratio across multiple queries with randomized trajectory orderings
>
> This formulation ensures that the final confidence captures both internal certainty and robustness under minor input perturbations, improving the reliability of modality-specific estimates. In preliminary experiments, we observed that this combined signal led to improved label accuracy compared to using either component in isolation.
>
> We will clarify this motivation further in the revised version.
>
> ---
> **References:**
>
> [1] Kwon, N., Di Palo, N., & Johns, E. (2024). *Language Models as Zero-Shot Trajectory Generators*. IEEE Robotics and Automation Letters.
>
> [2] Tu, S., Sun, J., Zhang, Q., Lan, X., & Zhao, D. (2025). *Online Preference-Based Reinforcement Learning with Self-Augmented Feedback from Large Language Model*. Proceedings of the 24th International Conference on Autonomous Agents and Multiagent Systems (AAMAS).
>
> [3] Geng, J., Cai, F., Wang, Y., Koeppl, H., Nakov, P., & Gurevych, I. (2023). *A Survey of Confidence Estimation and Calibration in Large Language Models*. arXiv preprint arXiv:2311.08298.

---

> > ### Comment · Reviewer_87iK · 2025-08-01
> >
> > Thank your for your detailed response. I recommend to add the *"detailed comparison of compute costs and efficiency trade-offs, presented in Tables 1 & 2"* to the paper as it is a useful addition and a natural question to ask as a reader.
> >
> > I have no further questions and will stick with my positive assessment. Thank you.

---

> > > ### Author Response · Authors · 2025-08-01
> > >
> > > Thank you for your time in reviewing our rebuttal! We will incorporate the compute cost and efficiency trade-off tables into the revised manuscript as recommended.

---

### Official Review · Reviewer_Cuxg · 2025-06-30

**Clarity:** 3
**Significance:** 2
**Originality:** 2
**Rating:** 5
**Confidence:** 3

**Summary:**

This paper introduces PRIMT, a foundation model-driven framework for preference-based reinforcement learning (PbRL) that addresses challenges in synthetic feedback quality, query ambiguity, and credit assignment. PRIMT employs a hierarchical neuro-symbolic fusion strategy to integrate multimodal feedback from large language models (LLMs) and vision-language models (VLMs). Experiments across 2 locomotion and 6 manipulation tasks validate the effectiveness of its components.

**Questions:**

1. How much did it cost to tune the gpt-4o api for the whole experiment?

**Ethical Concerns:**

["NO or VERY MINOR ethics concerns only"]

**Final Justification:**

Resolved issues:
1. Hard to tune the parameters.
2. Lack of hyperparameter analysis.

Unresolved issues:
N/A

**Quality:**

3

**Strengths And Weaknesses:**

Strengths:
1. The paper is well-organized and easy to follow.
2. The experiments are very adequate and comprehensively demonstrate the effectiveness of the algorithm.
3. Real-world experiment on Kinova Jaco robot enhances the paper’s credibility, demonstrating practical applicability beyond simulation.

Weaknesses
1. The method is overly complex and difficult to re-implement, and too much hyperparameters make it hard to tune the parameters.
2. Lack of hyperparameter analysis. Conducting more experiments on hyperparameter tuning or ablation studies could be beneficial.

---

> ### Author Rebuttal · Authors · 2025-07-29
>
> # Response to Reviewer Cuxg
>
> **Dear Reviewer Cuxg,**
>
> Thank you for your positive assessment of our presentation, contributions, and experiments, and for your insightful comments. We are glad to respond to your concerns below.
>
> ---
>
> ### **W1 & W2: Complexity and Hyperparameters**
>
> We acknowledge that PRIMT consists of multiple components. However, each component is **well-motivated by specific limitations** in existing PbRL approaches:
>
> - Multimodal feedback fusion addresses the complementary weaknesses of single-modality evaluation (as demonstrated in Appendix A)
> - Foresight generation tackles early-stage query ambiguity when random policies produce uniformly poor trajectories
> - Hindsight augmentation improves credit assignment through targeted counterfactual reasoning
>
> As shown in our ablation studies (Figure 3 and Section G.1), each component contributes meaningfully to performance improvements, validating our overall architecture. Our goal is to develop a general zero-shot PbRL framework by fully leveraging foundation models, and the multi-agent/component design reflects this objective.
>
> ---
> Regarding hyperparameters, we clarify that they fall into two distinct categories:
>
> **1. Backbone PbRL training parameters**
> PRIMT is designed to be plug-and-play with any PbRL backbone. In our experiments, we adopt the PEBBLE setup for reward model and policy learning (e.g., optimizer, learning rate, batch size, training schedule), as detailed in Appendix F.2 and F.3. These were **shared across all baselines and tasks without individual tuning**.
>
> **2. PRIMT-specific module parameters**
> The hyperparameters related to PRIMT modules were mostly **held constant across all tasks**:
> - The confidence balancing weight α in Eq. 5 was fixed at 0.5
> - PSL rule weights were uniformly set to [3, 1, 1, 1] for the agreement, conflict resolution, and indecision rules (Eqs. 6, 7, 8, 12)
> - The number of foresight and hindsight trajectories per round was fixed as described in Appendix F.2
>
> These parameters were not extensively tuned, and the consistent performance across diverse tasks spanning multiple benchmark suites suggests that PRIMT **does not heavily depend on extensive hyperparameter tuning to be effective**.
>
> That said, we fully agree that conducting more systematic hyperparameter analysis would be valuable for the community. While current resource constraints prevent extensive additional experiments, we acknowledge this as an exciting direction. In the Discussion section of the revised manuscript, we will highlight this as a promising direction for future research, such as sensitivity analysis of VLM/LLM agent quantities in crowd-checking and the impact of foresight/hindsight trajectory amount requirements on performance.
>
> For reproducibility, we have provided comprehensive implementation details in the appendix and plan to release our codebase upon acceptance to facilitate such investigations by the research community.
>
> We will revise the manuscript to better foreground these design motivations and hyperparameter grouping for clarity.
>
> ---
>
> ### **Q1: LLM Usage Cost**
>
> The average cost per task ranges from 79-124 USD across different benchmark environments, making training on all 8 tasks cost approximately $850 in total. We provide detailed cost breakdowns and trade-off analysis in our response to Reviewer r5ng (Tables 1 and 2) for additional context.
>
> Compared to FM-based baselines, PRIMT increases costs moderately (38-47%) while delivering substantial performance gains (19-117%), achieving 1.4-2.0× efficiency improvements. PRIMT also achieves comparable performance to expert human teachers (within 1-3%) while reducing human annotation costs by 92-95%.
>
> ---
>
>
> Thank you again for your constructive feedback and positive evaluation!

---

### Official Review · Reviewer_r5ng · 2025-07-02

**Clarity:** 3
**Significance:** 3
**Originality:** 3
**Rating:** 4
**Confidence:** 3

**Summary:**

The paper describes a preference-based RL technique where feedback is produced by VLMs and LLMs. LLMs can interpret trajectories, e.g. in the form of state-action pairs, but such descriptions can be incomplete and the LLM can be confused by some fine-grained spatial interactions. VLMs offer an alternative, but have their own limitations, e.g. in terms of recognising subtle temporal dynamics. Hence, the work's pursuit of a multimodal approach where feedback from both LLMs and VLMs (with help of a keyframe extraction method) is fused. Preference fusion is supported by the use of Probabilistic Soft Logic (PSL) helping to define rules for fusion (i.e. agreement, conflict resolution and indecision). The issues of low-quality trajectory pairs early in training and inaccurate state-action-level credit assignment (i.e. attributing preference differences to specific states / actions) are overcome with the help of foresight trajectory generation (using LLMs) and hindsight trajectory augmentation.

Results are generated for two locomotion and six manipulation tasks. PRIMT is compared against a VLM-based method, LLM-based methods, a naive multimodal approach, and the use of expert-designed reward functions (representing the upper bound, "prefGT"). gpt-4o is used as the FM backbone (section G.2 also compares to the use of the weaker gpt-4o-mini model). An ablation study supports the inclusion of the major components of the technique. Results show that PRIMT is competitive with the prefGT baseline and normally significantly outperforms other baselines.

**Questions:**

1. For PrefCLM, how many LLM agents were in the crowd?
2. G.6 can you quantify "superior performance" for this experiment?
3. Could you comment more on the LLM query cost? (beyond H.1). What if you simply compared techniques based on time or compute costs? Are there obvious ways to reduce the compute costs? (beyond cheaper/local models)

**Ethical Concerns:**

["NO or VERY MINOR ethics concerns only"]

**Final Justification:**

The authors have provided useful answers to my questions. I remain positive and will retain my rating at 4.

**Limitations:**

yes

**Quality:**

3

**Strengths And Weaknesses:**

Strengths: Interesting work that has produced significant improvements in performance.
Weaknesses: The FM usage represents a very significant amount of compute

---

> ### Author Rebuttal · Authors · 2025-07-28
>
> # Response to Reviewer r5ng
>
> **Dear Reviewer r5ng,**
>
> Thank you for your encouraging assessment of our work and for your thoughtful comments. We are grateful for the opportunity to respond to your questions as follows:
>
> ---
>
> ### **Q1. Crowd Size in PrefCLM**
>
> In our experiments, we followed the original setting in the source paper and used a crowd size of 10 for PrefCLM. We will add this detail, along with other implementation details of the baseline, in Section F.1 of the revised manuscript.
>
> ---
>
> ### **Q2. Quantifying “Superior Performance” in G.6**
>
> In the real-world Block Lifting and Block Stacking tasks, we evaluated each model over 10 randomized trials (e.g., varying initial block positions). PRIMT achieved 8/10 successful lifts and 6/10 successful stacks, whereas the PrefCLM baseline achieved 4/10 and 2/10 successes, respectively. This represents a doubling of the success rate compared to the baseline. We will include these success rates in Section G.6 to support the “superior performance” claim with concrete evidence.
>
> ---
>
> ### **Q3 & W1. FM Usage and Cost**
>
> We acknowledge that our method involves greater usage of foundation models (FMs) than prior work, due to its multi-agent nature. To provide a clearer picture of the resource requirements, we present a detailed **resource usage comparison** in **Table 1**, and the corresponding **cost-performance trade-offs** in **Table 2**.
>
> Note that we exclude PrefCLM from this comparison for fairness. This baseline assumes access to environment code and prompts LLMs to generate evaluation functions directly in code, thereby avoiding LLM-based preference queries and incurring minimal FM usage cost.
>
> We observe that compared to the **RL-VLM-F** and **Sa-LLM-F** baselines, **cost** and **training time** increased moderately (by **38–47%** and **30–69%**, respectively), while **performance gains** were substantial (**+19–117%**), resulting in **efficiency improvements of 2.0× and 1.4×**, respectively. This justifies the additional FM usage in PRIMT, given the performance benefits it enables.
>
> More importantly, compared to the performance achieved by collecting human feedback—represented by PrefGT where expert scripted teachers provide ground-truth preferences—PRIMT achieves **comparable performance** (within **1–3%**) while **reducing estimated human annotation costs by over 92%** ( based on ~$0.05–$0.10 per preference label for 20,000 queries on platforms such as Prolific and MTurk). This highlights PRIMT’s scalability and practicality as an alternative to expensive human-in-the-loop methods.
>
>  Overall, we believe PRIMT strikes a **good balance between performance and cost-effectiveness**, providing a practical path toward scalable preference learning.
>
> ---
>
> ### **Table 1: Resource Usage Comparison**
>
> | Method         | Usage Cost ($)     | Training Time (h)      |
> |----------------|-------------------------------|-------------------------|
> | **RL-VLM-F**   | 84.14 / 57.42 / 84.27         | 4.3 / 5.1 / 4.5         |
> | **Sa-LLM-F**   | 83.22 / 55.31 / 89.69         | 5.2 / 5.6 / 5.2         |
> | **PRIMT**      | 120.42 / 79.73 / 124.01       | 6.8 / 7.3 / 7.6         |
> | **Human**      | 1,000–2,000 (all envs)        | N/A                     |
>
> *Values across MetaWorld / ManiSkill / DMC, respectively*
>
> *FM cost estimated based on gpt-4o API pricing at the time of experiments*
>
> *Human cost estimated from 20,000 preference queries at $0.05–$0.10 per label on platforms like Prolific and MTurk*
>
> ---
>
> ### **Table 2: Cost-Performance Trade-off**
>
> | Baseline | **Cost** | **Time** | **Performance** | **Efficiency†** |
> |----------|----------|----------|-----------------|-----------------|
> | **vs RL-VLM-F** | **+43%/+39%/+47%** | **+58%/+43%/+69%** | **+95%/+117%/+68%** | **2.0×** |
> | **vs Sa-LLM-F** | **+45%/+44%/+38%** | **+31%/+30%/+46%** | **+32%/+109%/+19%** | **1.4×** |
> | **vs Human (PrefGT)** | **−92%/−95%/−92%** | **—/—/—** | **−1%/−3%/−2%** | **47×** |
>
> *Values across MetaWorld / ManiSkill / DMC, respectively*
>
> *Performance is measured using the final return from the learning curves presented in Figure 2 of the paper*
>
> *† Efficiency = Average performance gain / Average resource increase*
>
> ---
>
> Additionally, our ablation study in Section G.2 suggests that our method shows reasonable **robustness to cheaper foundation models**: GPT-4o-mini achieves 94% cost savings with only 16-26% performance reduction, resulting in 13× better cost-performance efficiency and suggesting potential for more accessible deployment.
>
> That being said, we fully acknowledge the importance of **reducing FM usage cost**. PRIMT already adopts several strategies to this end. For example, we extract **keyframes** from trajectories instead of passing full image sequences to the VLM. For foresight generation, we leverage **code-based** generation to synthesize multiple trajectories offline in a single batch, reducing LLM cost. We also implement **parallel querying** for VLM/LLM-based preference generation to reduce training time and wall-clock latency.
>
> In the revision, we will include a more detailed cost analysis and expand the discussion of optimization strategies to make our approach even more cost-effective and scalable for broader adoption.
>
> ---
>
> Thank you again for your time in reviewing our paper and rebuttal!

---

> > ### Comment · Reviewer_r5ng · 2025-08-04
> >
> > Thank you for your detailed response to my questions. I remain positive and have no further questions for you.

---

> > > ### Author Response · Authors · 2025-08-04
> > >
> > > Thank you very much for your follow-up and continued positive assessment! We greatly appreciate your time and feedback throughout the review process.

---

### Note · Authors · 2025-08-16

We thank the reviewers and ACs for their time and engagement during the review and rebuttal process. We are encouraged that all reviewers found our rebuttal satisfactorily addressed their questions and maintained their positive assessments. We will incorporate the additional clarifications (e.g., cost–efficiency analysis) and supporting results (e.g., inclusion of a new RL baseline and preliminary experiments on more complex tasks) into the revised manuscript.

---

### Decision · Program_Chairs · 2025-09-17

**Decision:**

Accept (oral)

**Comment:**

This paper tackles the problem of Preference-based Reinforcement Learning in developing the proposed system, PRIMT. This approach how to handle multi-modal feedback through a fusing feedback from a large language model and a vision language model. A neurosymbolic approach is used for this fusion. Next, the approach explores how to populate the PbRL replay buffer without assuming that the demonstrations are optimal -- an approach called "foresight" in this paper. In hindsight, the approach generates counterfacutals for learning as well. The approach is evaluated empirically and with ablations and a real-world evaluation on a Jaco robot. Reviewers felt the authors delivered on what they claimed as contributions. The paper was clear, the topic relevant, and the approach well-evaluated. However, there were concerns about how complex the model is -- perhaps with insufficient design and implementation details. As one reviewer summarized, the paper appears to be basically "`stacking heuristics` and informed guesswork." The appraoch also seems quite computationally expensive. The paper could be improved by providng a better understanding of what components of necessary vs. sufficient and why. Overall, the paper is solid and the real-world demonstration is a plus.